# Changes in local mineral homeostasis facilitate the formation of benign and malignant testicular microcalcifications

Ida Marie Boisen[1,2], Nadia Krarup Knudsen[1], John E Nielsen[3], Ireen Kooij[1], Mathilde Louise Bagger[1], Jovanna Kaludjerovic[4], Peter O'Shaughnessy[5], Peter W Andrews[6], Noriko Ide[4], Birgitte G Toft[7], Anders Juul[3], Arnela Mehmedbasic[1], Anne Jørgensen[1], Lee B Smith[8], Richard Norman[9], Ewa Rajpert-De Meyts[3], Beate Lanske[4], Martin Blomberg Jensen[1,10]*

[1]Division of Translational Endocrinology, Department of Endocrinology and Internal Medicine, University Hospital Copenhagen, Herlev-Gentofte, Herlev, Denmark; [2]Novo Nordisk Foundation Center for Basic Metabolic Research, University of Copenhagen, Copenhagen N, Denmark; [3]Department of Growth and Reproduction, Rigshospitalet, University of Copenhagen, Copenhagen, Denmark; [4]Division of Bone and Mineral Research, Harvard School of Dental Medicine/Harvard Medical School, Harvard University, Boston, United States; [5]School of Biodiversity, One Health & Veterinary Medicine, University of Glasgow, Glasgow, United Kingdom; [6]Centre for Stem Cell Biology, Department of Biomedical Science, University of Sheffield, Western Bank, Sheffield, United Kingdom; [7]Department of Pathology, Rigshospitalet, Copenhagen, Denmark; [8]MRC Centre for Reproductive Health, University of Edinburgh, The Queen's Medical Research Institute, Edinburgh, United Kingdom; [9]Department of Urology, Dalhousie University, Halifax, Canada; [10]Department of Clinical Medicine, Copenhagen University Hospital, Copenhagen, Denmark

*For correspondence:
blombergjensen@gmail.com

## eLife Assessment

This **valuable** study reports the link between a disruption in testicular mineral (phosphate) homeostasis, FGF23 expression, and Sertoli cell dysfunction. The data supporting the conclusion are **solid**. This work will be of interest to biomedical researchers working on testis biology and male infertility. The assessment is based on the editors' critical evaluation of the authors' responses.

**Abstract** Testicular microcalcifications consist of hydroxyapatite and have been associated with an increased risk of testicular germ cell tumors (TGCTs) but are also found in benign cases such as loss-of-function variants in the phosphate transporter *SLC34A2*. Here, we show that fibroblast growth factor 23 (FGF23), a regulator of phosphate homeostasis, is expressed in testicular germ cell neoplasia in situ (GCNIS), embryonal carcinoma (EC), and human embryonic stem cells. FGF23 is not glycosylated in TGCTs and therefore cleaved into a C-terminal fragment which competitively antagonizes full-length FGF23. Here, *Fgf23* knockout mice presented with marked calcifications in the epididymis, spermatogenic arrest, and focally germ cells expressing the osteoblast marker Osteocalcin (gene name: *Bglap*, protein name). Moreover, the frequent testicular microcalcifications in mice with no functional androgen receptor and lack of circulating gonadotropins are associated with lower *Slc34a2* and higher *Bglap*/*Slc34a1* (protein name: NPT2a) expression compared with wild-type mice. In accordance, human testicular specimens with microcalcifications also have lower

*SLC34A2* and a subpopulation of germ cells express phosphate transporter NPT2a, Osteocalcin, and RUNX2 highlighting aberrant local phosphate handling and expression of bone-specific proteins. Mineral disturbance in vitro using calcium or phosphate treatment induced deposition of calcium phosphate in a spermatogonial cell line and this effect was fully rescued by the mineralization inhibitor pyrophosphate. In conclusion, testicular microcalcifications arise secondary to local alterations in mineral homeostasis, which in combination with impaired Sertoli cell function and reduced levels of mineralization inhibitors due to high alkaline phosphatase activity in GCNIS and TGCTs facilitate osteogenic-like differentiation of testicular cells and deposition of hydroxyapatite.

## Introduction

Testicular microlithiasis in infertile men is associated with an increased risk of testicular germ cell tumors (TGCTs) (*Tan et al., 2010*; *Pedersen et al., 2016*; *Leblanc et al., 2018*; *Barbonetti et al., 2019*). However, both ultrasonic and histological microcalcifications are also frequent findings in the testis of asymptomatic men and the etiology remains largely unknown in both malignant and benign cases. The histological microcalcifications have been shown by Raman spectroscopy to consist of hydroxyapatite (*De Jong et al., 2004*; *Smith et al., 1999*), which is normally found exclusively in the skeleton and requires the presence of cells with osteoblast-like characteristics (*Kirsch, 2006*; *Boström et al., 1995*). Extra-skeletal mineralization is often caused by an imbalance in promoters and inhibitors of biomineralization (*Kirsch, 2006*; *Rowe, 2012*; *Martin and Quarles, 2012*; *Shanahan et al., 2011*). Tissue mineralization is tightly controlled by small integrin-binding ligand N-linked glycoproteins (SIBLINGs) such as osteopontin (OPN) and dentin matrix acidic phosphoprotein 1 (DMP1) that regulate mineralization in concert with osteocalcin and runt-related transcription factor 2 (RUNX2). Pyrophosphate (PPi) generated by ectonucleotide pyrophosphatase/phosphodiesterase 1 (ENPP1) is an essential suppressor of hydroxyapatite deposition and alkaline phosphatase (ALP) inactivates PPi (*Hessle et al., 2002*). ALP abundance is thus a central factor for local phosphate homeostasis, and during osteogenic differentiation increased ALP activity is a hallmark of the bone-like characteristics of the cells in other extra-skeletal tissue such as blood vessels. In this paper, ALP is used as an acronym for all four isozymes of ALP as all can be found in TGCTs (*Hofmann et al., 1989*). In addition, accumulating evidence suggests that fibroblast growth factor 23 (FGF23) and its receptor Klotho (gene name: *KL*), the phosphate transport proteins solute carrier family NPT2-a and -b (gene names: *SLC34A-1* and *-2*, respectively), and vitamin D metabolizing enzymes are essential for maintaining stable mineral homeostasis and that imbalance in their activity may lead to extra-skeletal calcifications (*Yuan et al., 2014*; *Lieben et al., 2012*; *Golub, 2009*; *Komori et al., 1997*; *Ducy et al., 1996*). Interestingly, loss-of-function variants in the *GALNT3* gene (protein name: GalNAc-T3) and hence a lack of degradation-protective glycosylation of FGF23 leads to systemic hyperphosphatemia and severe testicular microcalcifications (*Garringer et al., 2007*; *Campagnoli et al., 2006*; *Kato et al., 2006*). Intratesticular phosphate levels seem to be essential in the formation of testicular microlithiasis as loss-of-function variants in the main testicular phosphate transporter *SLC34A2* also lead to severe testicular microcalcifications without affecting systemic phosphate levels (*Corut et al., 2006*). Testicular microlithiasis is also found in patients with loss-of-function variants in the gene encoding ATP binding cassette subfamily C member 6 (*ABCC6*), which results in abnormal PPi metabolism and thus less inhibition of mineralization (*Vanakker et al., 2006*). These studies show that particularly local and potentially systemic changes in phosphate homeostasis induce testicular microcalcifications. Therefore, we hypothesized that the etiology of testicular microcalcifications involves disturbances in intratesticular mineral homeostasis.

TGCTs originate from a common progenitor germ cell neoplasia in situ (GCNIS), which is characterized by the expression of pluripotency factors such as octamer-binding transcription factor 4 (gene names: *POU5F1*, protein name: OCT4), NANOG, and high ALP activity (*Hofmann et al., 1989*; *Skakkebaek, 1972*; *Rajpert-De Meyts et al., 2004*; *Hart et al., 2005*; *Jacobsen and Nørgaard-pedersen, 1984*). GCNIS cells undergo malignant transformation and form either a seminoma that retains germ cell characteristics or a non-seminoma that may contain embryonal carcinoma (EC) resembling human embryonic stem cells (hESCs), teratoma, yolk sac tumor, choriocarcinoma, or a combination of these (*Rodriguez et al., 2003*). During the malignant transformation of GCNIS into invasive TGCTs, duplication of chromosome 12p (often as isochromosome 12p) is the most consistent chromosomal

aberration and may be involved in the dedifferentiation from GCNIS to EC (*Rodriguez et al., 2003*). In healthy individuals, FGF23 is produced exclusively in osteocytes, and the location of the gene on chromosome 12p may be important for its presence and role in TGCTs. Full-length FGF23 increases phosphate excretion by decreasing the luminal expression of NPT2a in the kidney and by lowering the concentration of active vitamin D via increased expression of the inactivating enzyme CYP24A1 and reduced expression of the activating enzyme CYP27B1 (*Shimada et al., 2004*). We have previously shown that the local metabolism of vitamin D disappeared during the malignant transformation of GCNIS to EC and that FGF23 was expressed in an EC-derived cell line (*Blomberg Jensen et al., 2012*). *O*-glycosylation by GalNAc-T3 protects FGF23 from rapid degradation, and full-length FGF23 activates a receptor complex consisting of Klotho heterodimerization with FGF receptor 1 (FGFR1). However, the cleaved C-terminal fragment (cFGF23) can bind and serve as a competitive antagonist for intact FGF23 (iFGF23), and iFGF23 can also induce Klotho-independent effects (*Goetz et al., 2010*). Loss-of-function variants in the genes *FGF23*, *GALNT3*, or *KL* lead to tumoral calcinosis partly through osteogenic-like differentiation of mesenchymal-derived cells due to altered mineral homeostasis and vitamin D metabolism (*Farrow et al., 2011*). Here, we investigated whether ectopic FGF23 production and abundant ALP activity in TGCTs alter testicular mineral homeostasis, which alone or in combination with impaired Sertoli cell function may facilitate osteogenic-like differentiation and thus be responsible for the frequent deposition of hydroxyapatite in TGCTs and dysgenetic testes. To address this question, we investigated the expression of mineral regulators and bone factors in human tissues containing TGCT-associated calcifications, *Fgf23* knockout (*Fgf23*$^{-/-}$) mice, hypogonadal (no gonadotropins) mice, androgen-insensitive mice with microcalcifications, mice with microcalcifications due to Sertoli cell ablation, mice xenografted with a TGCT-derived cell line, mice treated with a high phosphate diet, mineral deposition in a mouse spermatogonial cell line (GC1), and an ex vivo mouse testis culture model.

## Results

### Testicular microcalcifications are associated with the presence of renal phosphate transporters and bone markers in human testis

Ultrasonic testicular microlithiasis is often observed adjacent to TGCTs but can also be found in testes without TGCTs or GCNIS (*Figure 1A*). Ultrasonic testicular microlithiasis is not always associated with histological microcalcifications but intratesticular mineral depositions can be confirmed by von Kossa or alizarin red staining (*Figure 1A*), and Raman spectroscopy has previously demonstrated that testicular microcalcifications consist of hydroxyapatite (*De Jong et al., 2004*). In GCNIS-containing tubules adjacent to microcalcifications, we found that the renal phosphate transporter NPT2a and FGF23 were ectopically expressed. In contrast, Klotho was expressed in the cytoplasm of the germ cells in normal tubules, with or without adjacent microcalcifications, but not in GCNIS. FGFR1 expression was found in both GCNIS and normal tubules (*Figure 1B*). The expression of the bone marker Osteocalcin was present in tissue with GCNIS and adjacent to the microcalcifications but not the normal tubules with full spermatogenesis (*Figure 1B* and *Table 1*). The microcalcifications vary in size from small calcifications focally in the center of seminiferous tubules to occupying most of the intraluminal area (*Figure 1C*). In sporadic severe cases there was intratesticular bone formation (*Figure 1D*). We investigated the presence of several bone factors in the cells adjacent to the microcalcifications. The early osteoblast marker RUNX2 was not expressed in testes with normal spermatogenesis. However, occasionally, when microcalcifications or bone formation were detected, a small fraction of the germ cells in the seminiferous tubules had cytoplasmic staining of RUNX2 supporting a bone-like phenotype of some germ cells (*Figure 1*, *Figure 1—figure supplement 1*, and *Table 1*). The shift in expression of the phosphate transporters from NPT2b to NPT2a and the presence of bone factors in testis with microcalcifications were also evident at the transcriptional level. *SLC34A1* was expressed at a very low level in normal testis but was highly expressed in 25% of the specimens containing GCNIS. In contrast, the phosphate transporter *SLC34A2* was abundantly expressed in the germ cells from normal testis but the expression was lower in specimens containing GCNIS (*Figure 1E*).

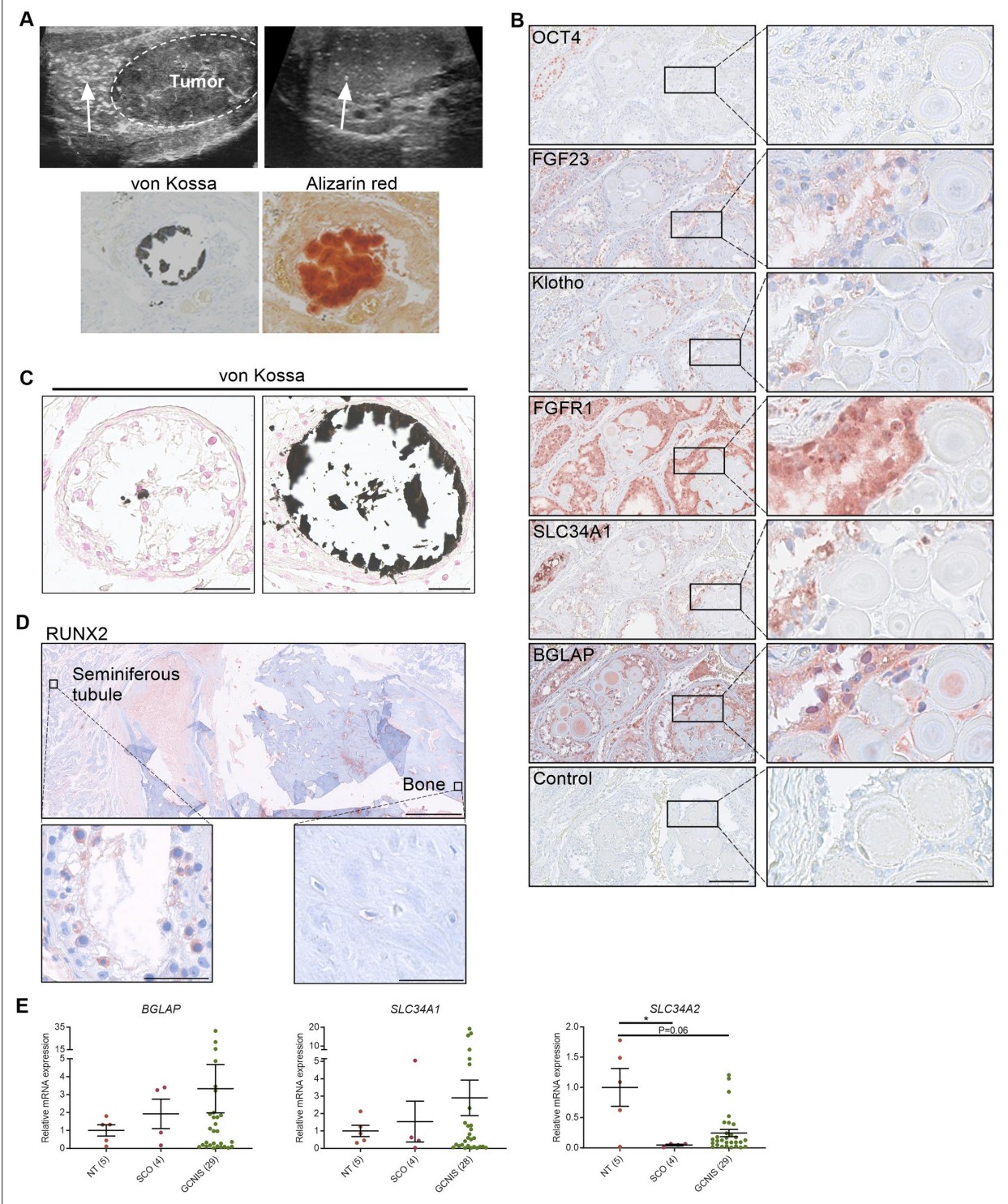

**Figure 1.** Deposition of hydroxyapatite is linked with the induction of bone markers and changes in the expression of phosphate transporters. (**A**) Ultrasonic detection of testicular microlithiasis adjacent to a tumor (left) or in an otherwise healthy testis (right). Arrows indicate testicular microlithiasis. Staining of calcifications by von Kossa and alizarin red in human testis. (**B**) IHC staining of proteins potentially involved in the formation of testicular microcalcifications adjacent to microcalcifications in human testis. OCT4: marker of germ cell neoplasia in situ (GCNIS) cells. Control: without primary antibody. Scale bars correspond to 125 µm (left) and 25 µm (right). (**C**) Mild (left) and moderate (right) microcalcifications in human testis stained with von Kossa. Scale bars correspond to 50 µm. (**D**) Severe bone formation in testis from a man with no known history of local trauma or other predisposing factors of ectopic ossification. IHC staining with RUNX2. Scale bars correspond to 1.25 mm (upper), 50 µm (lower left), and 100 µm (lower

*Figure 1 continued on next page*

*Figure 1 continued*

right). (**E**) Expression levels of the bone marker (*BGLAP*) and phosphate transporters (*SLC34A1* and *SLC34A2*) in NT (normal testis), SCO (Sertoli-cell-only) pattern, and GCNIS. Data are presented as mean ± SEM. *p<0.05. ANOVA on log-transformed data was used as the statistical test.

The online version of this article includes the following source data and figure supplement(s) for figure 1:

**Source data 1.** Numerical data used to generate the figure.

**Figure supplement 1.** Deposition of hydroxyapatite and bone markers.

## FGF23 is produced by human TGCTs and embryonic stem cells

FGF23, a potent regulator of phosphate homeostasis, is expressed in the EC-derived cell line NTera2 (*Blomberg Jensen et al., 2012*), and the *FGF23* gene is located on chromosome 12p which is duplicated in TGCTs and could be associated with the formation of testicular microcalcifications. Thus, expression of *FGF23* was investigated in human testicular specimens containing 'normal testis' (adjacent tissue to TGCT/GCNIS without presence of malignant cells), Sertoli-cell-only pattern, GCNIS, seminoma, and EC with or without teratoma components. The transcriptional level of *FGF23* was highest in EC components, significantly higher compared with normal testis, and was positively correlated with the expression of the pluripotency factor *NANOG* but not significantly with *POU5F1* (r=0.813, p=0.049, and r=0.780, p=0.068, respectively). Two EC specimens with polyembryoma components (the most dedifferentiated structure resembling early embryonic formation) had several fold higher levels of *FGF23* (*Figure 2A and B*). A higher expression of *FGF23* in GCNIS compared with normal testis was also confirmed in microdissected GCNIS cells, while fetal gonocytes and hESCs also expressed *FGF23* (*Figure 2C*). The high expression of *FGF23* in hESC and EC was validated by a ~30-fold higher expression in different strains of hESC cells and the EC-derived cell line NTera2 compared with normal testis (*Figure 2D*). In contrast, normal testicular tissue had a high expression of the FGF23 receptor Klotho (*KL*), whereas the expression was low in GCNIS cells and hESC (*Figure 2C*). IHC confirmed the marked expression of FGF23 solely in OCT4-positive GCNIS cells that also expressed FGFR1, while Klotho was exclusively expressed in normal germ cells and not found in any of the OCT4-positive cancer cells including seminoma and EC (*Figure 2E* and *Table 1*). FGF23 was also expressed in the EC cells characterized by concomitant OCT4 expression but was virtually absent in seminoma cells despite their marked OCT4 expression suggesting that the chromosome 12p duplication (occurring in both seminoma and EC) alone is not sufficient to maintain FGF23 production. Western blot confirmed the presence of FGF23 with two bands corresponding to the sizes of cFGF23 and iFGF23 in GCNIS and EC. cFGF23 was also expressed in NTera2 cells, and neither cFGF23 nor iFGF23 were detected in normal testis or the seminoma-derived cell line

**Table 1.** Immunohistochemical expression of selected proteins in human specimens.

| | n | FGF23 | Klotho | FGFR1 | Osteocalcin | RUNX2 | OPN |
|---|---|---|---|---|---|---|---|
| Normal testis | 7–15 | | ++to +++ | ++to +++ | | | + |
| Germ cell carcinoma in situ | 6–18 | +++ | to +/- | ++to +++ | + | +++ | + |
| Hyalinized/calcified testis | 5–14 | to + | +/to + | ++ | +++ | +++ | +++ |
| Classical seminoma | 4–15 | to + | | ++ | | | |
| Anaplastic seminoma | 2 | ++ | +/ | ++ | NA | NA | NA |
| Embryonal carcinoma | 4–13 | ++to +++ | to +/- | ++ | +/to ++ | +/to + | +/to + |
| Teratoma | 3–6 | to + | +/to ++ | ++ | +/to ++ | +/to + | +/to + |
| Fetal testis | 3 | to +/- | | ++ | NA | NA | NA |

- Not expressed.
+/- Barely detectable in few cells.
+ Expressed in few cells.
++ Markedly expressed in some cells or expressed in most cells.
+++ High expression in most cells.
N.A. Not available.

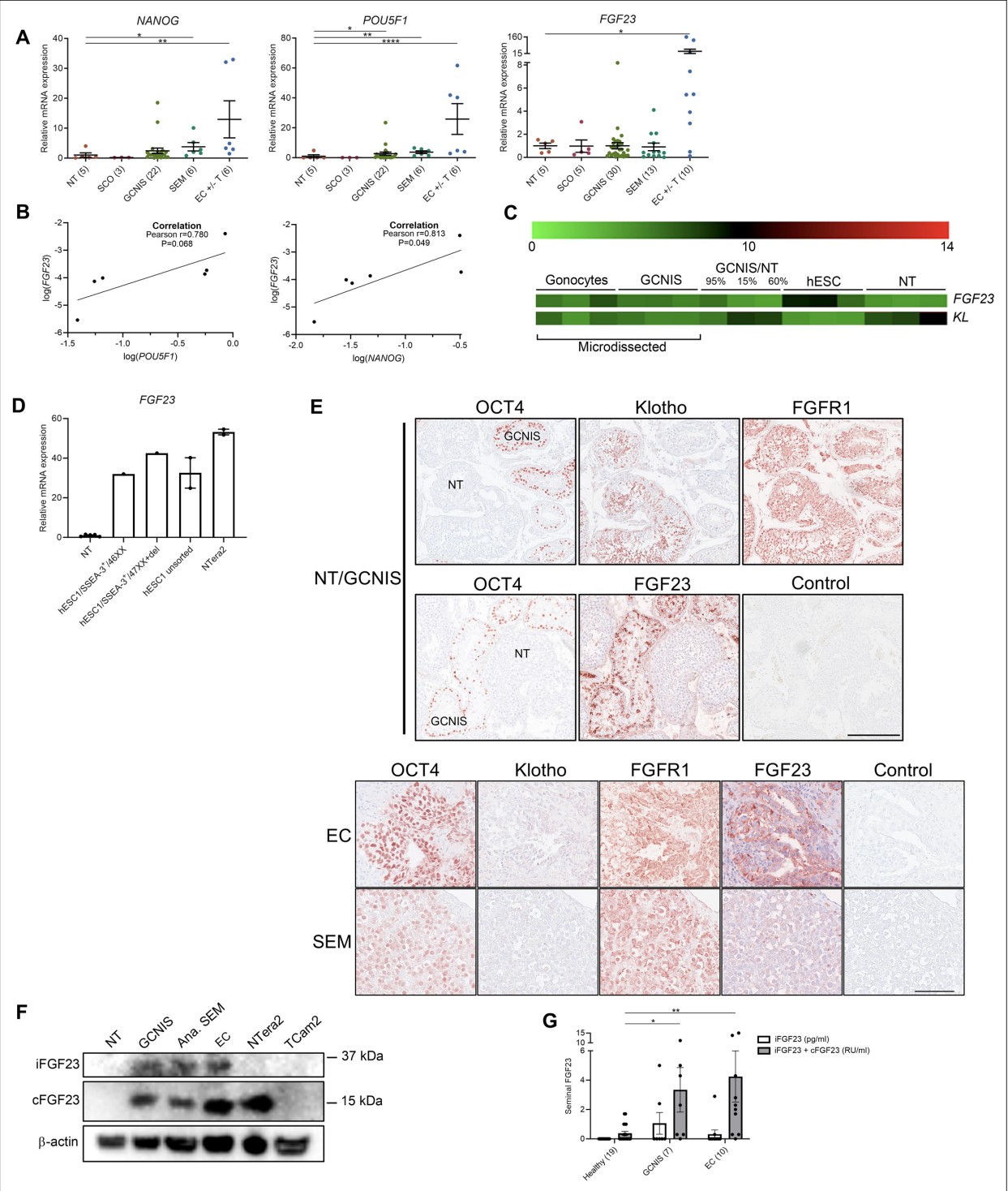

**Figure 2.** Ectopic expression of FGF23 in human testicular germ cell tumors and embryonic stem cells. (**A**) Expression levels of the pluripotency factors *NANOG, POU5F1*, and *FGF23* in normal testis (NT), Sertoli-cell-only (SCO) pattern, germ cell neoplasia in situ (GCNIS), seminoma (SEM), and EC +/- teratoma (**T**). ANOVA with Dunnett's multiple comparisons test on log-transformed data was used as the statistical test. (**B**) Correlations between gene expression of *FGF23* and *POU5F1* or *NANOG* in EC +/- T were tested with Pearson correlation tests. (**C**) Microarray showing expression levels of *FGF23* and *KL* in microdissected fetal gonocytes and GCNIS compared with testicular tissue samples with varying percentages of GCNIS, hESC, and NT. (**D**) Expression level of *FGF23* in NT, hESC, and NTera2 cells (n=1–5). (**E**) IHC staining of GCNIS with adjacent NT, EC, and SEM. Scale bars correspond to 250 µm (upper panels) and 100 µm (lower panels). (**F**) Western blot of iFGF23 and cFGF23 in NT, GCNIS, anaplastic SEM, EC, NTera2 cells, and TCam2

*Figure 2 continued on next page*

*Figure 2 continued*

cells. (**G**) Levels of seminal iFGF23 or total FGF23 (iFGF23+cFGF23) in healthy men or men with GCNIS or EC. Parentheses on the x-axis indicate sample size. Kruskal-Wallis test was used as the statistical test. Data are presented as mean ± SEM. *p<0.05, **p<0.01, ****p<0.0001.

The online version of this article includes the following source data for figure 2:

**Source data 1.** Numerical data used to generate the figure.

**Source data 2.** Original files for western blot analysis displayed in *Figure 2F*.

**Source data 3.** Original western blots for *Figure 2F*, indicating the relevant bands.

TCam2 (*Figure 2F*). cFGF23 and iFGF23 were expressed in an anaplastic seminoma sample, which is considered a rare intermediate between seminoma and EC due to the high proliferative index and morphological resemblance with EC (*Figure 2F* and *Table 1*). Production and release of FGF23 in GCNIS and EC cells were examined by measuring the total FGF23 (iFGF23+cFGF23) and cFGF23 concentrations in seminal fluid from healthy men and testicular tumor patients. Healthy men had undetectable iFGF23 levels in the seminal fluid, and a few (8/19) men had detectable cFGF23. A few men with GCNIS (2/7) or EC (2/10) had detectable levels of iFGF23 in the seminal fluid whereas total FGF23 were detected in the majority (9/10) of men with EC and their total FGF23 concentration in seminal fluid was significantly higher compared with healthy men (*Figure 2G*).

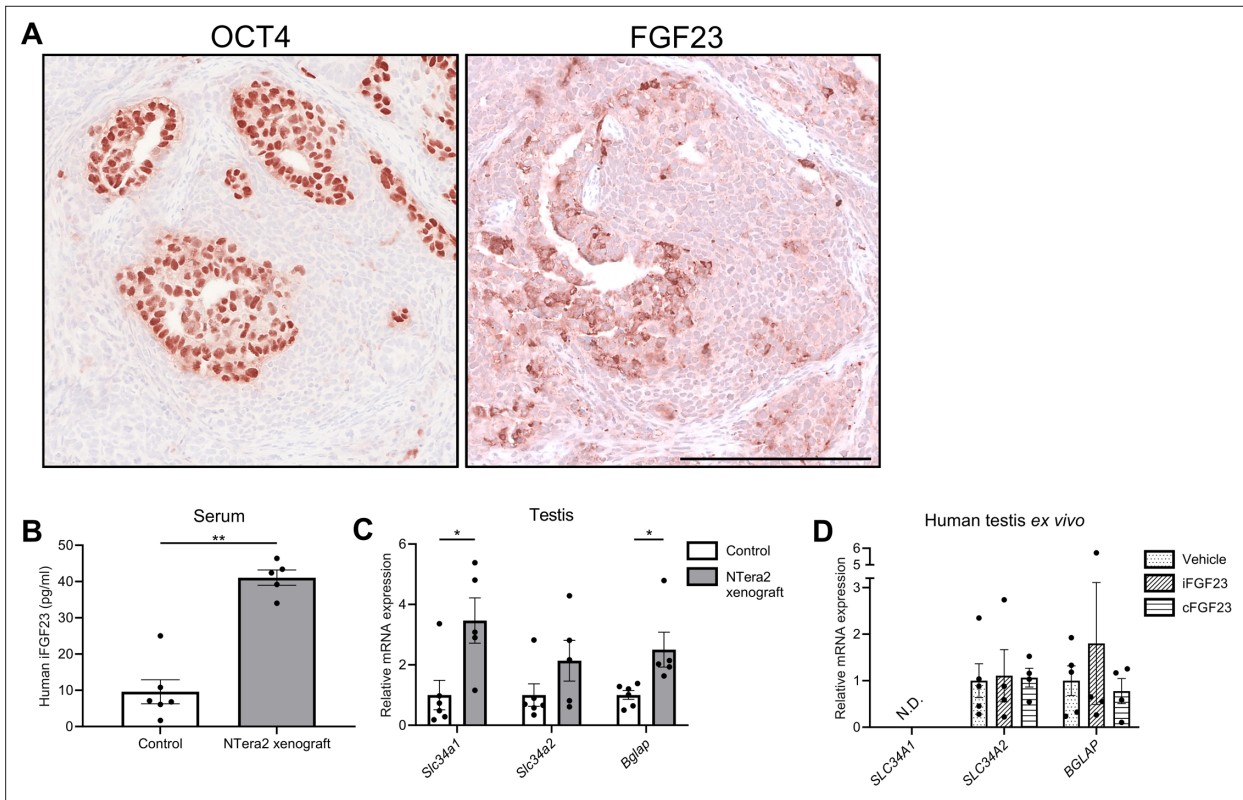

**Figure 3.** Change in expression of phosphate transporters and a bone marker in testis of FGF23-producing NTera2 cell tumors in xenografted mice. (**A**) OCT4 and FGF23 expression in the tumor on the flank of the nude mice. The scale bar corresponds to 250 µm. (**B**) Serum levels of human iFGF23 in control or NTera2 cell inoculated nude mice (n=5–6). A two-sided Student's t-test was used as the statistical test. (**C**) Expression level of the phosphate transporters *Slc34a1* and *Slc34a2* and the bone marker *Bglap* in testis of control or NTera2 cell xenografted nude mice (n=5–6). Two-sided Student's t-tests with Holm-Šídák correction were used as the statistical tests. (**D**) Effects of iFGF23 (50 ng/ml) or cFGF23 (200 ng/ml) after 24 hr in human ex vivo cultures of testis tissue without germ cell neoplasia in situ (GCNIS). ANOVAs with Dunnett's multiple comparisons test on log-transformed data were used as the statistical tests. Data are presented as mean ± SEM. *p<0.05, **p<0.01.

The online version of this article includes the following source data for figure 3:

**Source data 1.** Numerical data used to generate the figure.

## Testicular phosphate homeostasis in NTera2 xenograft mice is altered by FGF23-independent actions

In NTera2 xenograft tumors, FGF23 was expressed in only 5–10% of tumor cells, which shows that the phenotype of the NTera2 cells changes in vivo. This could be caused in part by somatic differentiation of the FGF23-negative fraction comprising 90–95% of the tumors, as also illustrated by the similar expression pattern of OCT4 within the tumor (*Figure 3A*). Human iFGF23 in serum from nude mice with tumors was fourfold higher compared with control nude mice (*Figure 3B*) and thus much lower than the EC tumor samples. The presence of iFGF23 in serum and the undetectable levels of iFGF23 in NTera2 cells grown in vitro suggests that NTera2 cells xenografted into nude mice undergo somatic differentiation in response to host stimuli and facilitate GalNAc-T3-mediated glycosylation of iFGF23 to prevent its cleavage (*Figure 2F*). Interestingly, NTera2 tumor-bearing mice had significantly higher *Slc34a1* and bone gamma-carboxyglutamic acid-containing protein (*Bglap*) expression in testis compared with nude mice without tumors but no change in *Slc34a2*, suggesting that the effect induced by the tumor was cFGF23 or FGF23 independent (*Figure 3C*). Exogenous treatment with iFGF23 or cFGF23 in human testis specimens cultured ex vivo for 24 hr did not change the expression of phosphate transporters or *BGLAP* (*Figure 3D*), indicating that FGF23 alone is not responsible for the change in expression of these genes but may be caused by other factors released or induced by the tumors.

## Mineralization in testis and epididymis of *Fgf23*^-/- mice and in mice treated with a high phosphate diet

*Fgf23*^-/- mice had elevated serum concentrations of calcium and phosphate, smaller testes, and often spermatogenic arrest at the spermatocyte stage compared with control mice (*Figure 4A* and data not shown). Testicular evaluation showed no deposition of hydroxyapatite, but DMP1, OPN, and Osteocalcin were expressed in the cytoplasm of some germ cells in up to 15% of the tubules in *Fgf23*^-/- mice and more rarely in WT mice (*Figure 4B and C*). To distinguish the effects of FGF23 loss from high phosphate alone, WT mice received a high phosphate diet, which showed that short-term high serum phosphate did not induce testicular microlithiasis or induce any testicular changes in the expression of phosphate transporters or *Bgalp* (*Figure 4D*). A more severe phenotype was observed in cauda epididymis with marked mineralization in 25% of the *Fgf23*^-/- mice and none of the WT mice. DMP1, OPN, and Osteocalcin were highly expressed in the epididymis of *Fgf23*^-/- mice (*Figure 4E and F*), while DMP1 and Osteocalcin were not detected in the epididymal lumen of WT mice. OPN was expressed in the head of sperm in both WT and *Fgf23*^-/- mice, although it was expressed exclusively in the luminal part in the calcified section of the epididymis of *Fgf23*^-/- mice.

## Testicular microcalcifications in hypogonadal, androgen receptor knockout, and Sertoli cell-ablated mice

Testicular microcalcifications are not exclusively found in mice with knockout of phosphate transport regulators but also in ~30% of hypogonadal (*hpg*) mice that lack circulating gonadotropins (follicle-stimulating hormone and luteinizing hormone). The proportion of mice with testicular microcalcifications increased to ~94% in mice with concomitant global loss of the androgen receptor (*AR*) (*hpg*. ARKO) (*O'Shaughnessy et al., 2009*) (schematically illustrated in *Figure 5A*). Sertoli cell-specific ablation of *AR* in *hpg* mice (*hpg*.SCARKO) did not augment the effect on microcalcifications in the *hpg* mice, indicating a role of AR in peritubular cells (the only other cell type in the testis that express AR) or AR-expressing cells outside the testis for the formation of microcalcifications. Thus, suggesting that somatic cells are important for the bone-like trans-differentiation in mice. Noteworthy, a shift in the expression of phosphate transporters was observed in the testis of *hpg*, *hpg*.SCARKO, and *hpg*.ARKO mice which had higher expression of *Slc34a1* and lower expression of *Slc20a1* and *Slc34a2* compared with WT mice. Thus, supporting the hypothesis that local phosphate homeostasis is involved in the formation of microcalcifications in accordance with the observations in human tissues with microcalcifications. There were no significant differences in *Slc20a1*, *Slc34a2*, and *Slc34a1* expression between *hpg*, *hpg*.SCARKO, or *hpg*.ARKO mice despite the higher proportion of mice with microcalcifications in the *hpg*.ARKO group. However, *Bglap* was detected in testis from all *hpg*.ARKO and most *hpg*. SCARKO mice, suggesting that *Bglap* is expressed in mice with disturbed Sertoli cell maturation with the highest expression found in *hpg*.ARKO mice which had a 94% prevalence of microcalcifications

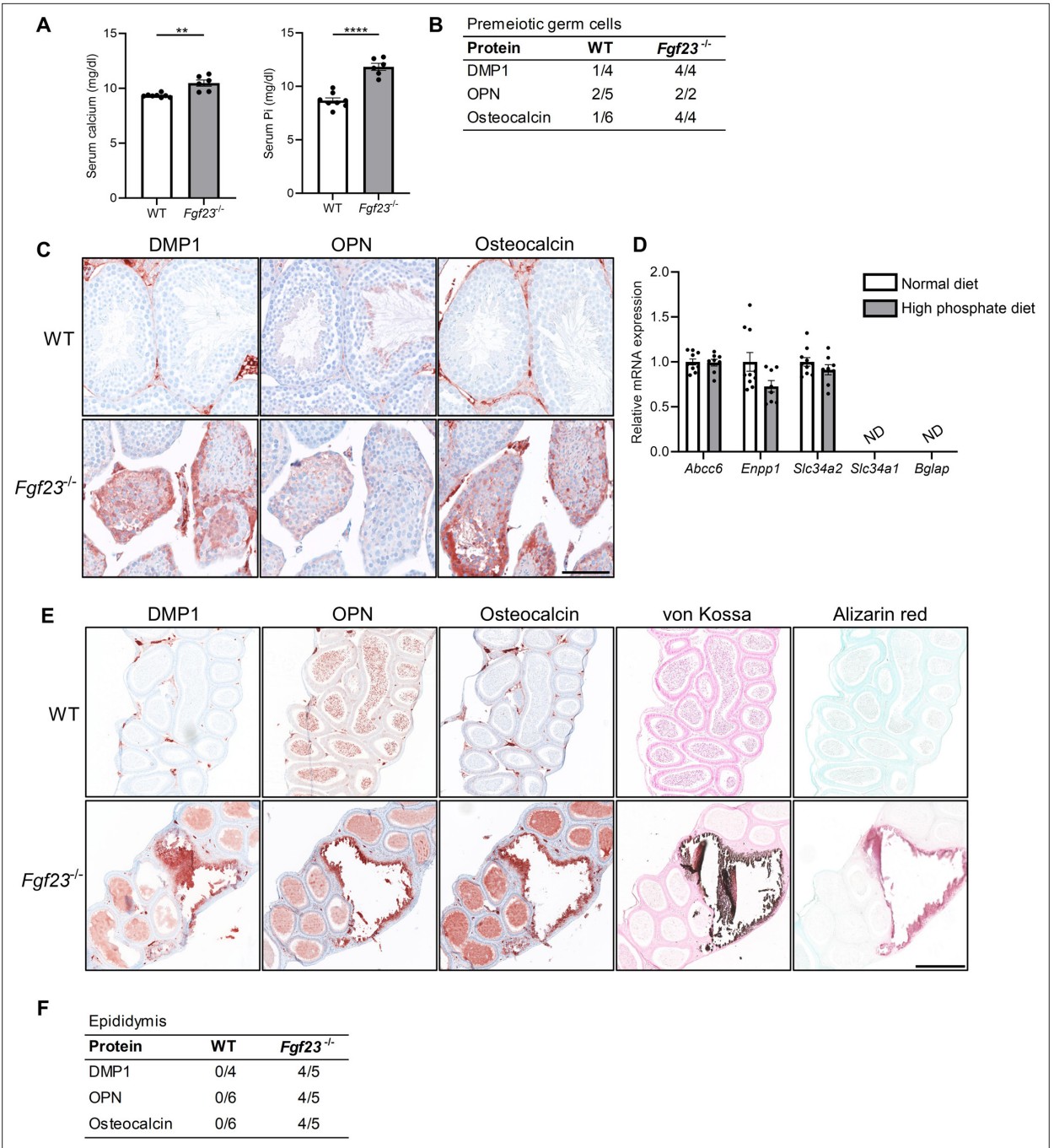

**Figure 4.** Mineralization in reproductive organs of *Fgf23* knockout mice. (**A**) Serum levels of calcium and inorganic phosphate (Pi) in wild-type (WT) and *Fgf23-/-* mice. Mann-Whitney (serum calcium) or two-sided Student's t-test (serum Pi) were used as the statistical tests. (**B**) Table of bone marker expression observed in premeiotic germ cells in WT and *Fgf23-/-* mice. (**C**) IHC staining of bone markers and vitamin D processing enzymes in WT and *Fgf23-/-* mice. Scale bar corresponds to 100 μm. (**D**) Expression level of the calcification regulators *Abcc6* and *Enpp1*, the phosphate transporters *Slc34a2* and *Slc34a1*, and the bone marker *Bglap* in testis of mice that received normal or high phosphate diet (n=8–10). Two-sided Student's t-tests with Holm-Šídák correction were used as the statistical tests. (**E**) IHC staining of bone markers and staining of minerals with von Kossa and alizarin red in epididymis of WT and *Fgf23-/-* mice. The scale bar corresponds to 250 μm. (**F**) Table of bone marker expression observed in the epididymis of WT and *Fgf23-/-* mice. Data are presented as mean ± SEM. **p<0.01 and ****p<0.0001.

The online version of this article includes the following source data for figure 4:

**Source data 1.** Numerical data used to generate the figure.

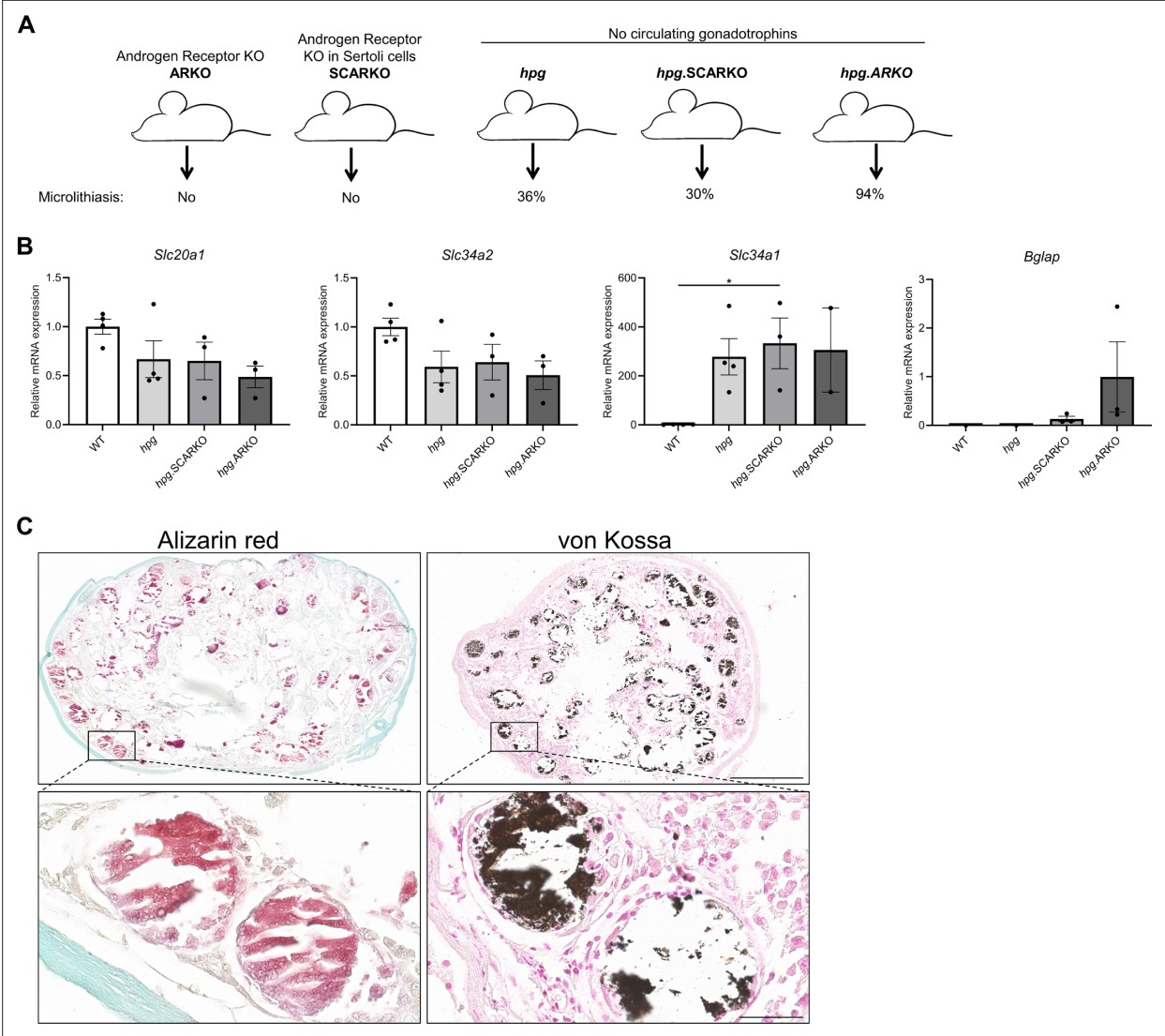

**Figure 5.** Testicular microcalcifications in hypogonadal and Sertoli cells-ablated mice. (**A**) Schematic representation of the prevalence of microlithiasis in ARKO, SCARKO, *hpg*, *hpg*.SCARKO, and *hpg*.ARKO mice models. (**B**) Expression levels of *Slc20a1*, *Slc34a2*, *Slc34a1*, and *Bglap* in testis of *hpg*, *hpg*. SCARKO, and *hpg*.ARKO mice models (n=2–4). Data are presented as mean ± SEM. *p<0.05. ANOVAs with Dunnett's multiple comparisons test were used as the statistical tests. (**C**) Microcalcifications in Sertoli cell-ablated mice. Mineral staining with alizarin red or von Kossa. Scale bars correspond to 500 μm (upper pictures) and 50 μm (lower pictures).

The online version of this article includes the following source data for figure 5:

**Source data 1.** Numerical data used to generate the figure.

(*Figure 5B*). Ablation of Sertoli cells at postnatal day 18 induced substantial microcalcifications in the adult mice (80 days of age) as shown by alizarin red and von Kossa staining (*Figure 5C*). The microcalcifications persisted when assessed in mice 1 year of age (data not shown). The timing of ablation seemed important because Sertoli cell ablation in adulthood (after puberty) did not cause microcalcifications when evaluated after 90 days but was detectable 1 year later (data not shown). This shows that immature prepubertal cells are more prone to form microcalcifications. The formation of microcalcifications depends on the presence of germ cells with stem cell potential as ablation of Sertoli cells during embryonic development, which causes loss of germ cells, did not result in testicular microcalcifications.

## ALPs in TGCTs and effects of calcium, phosphate, and PPi on mineralization in a spermatogonial cell line

ALP degrades the potent inhibitor of mineralization PPi and is highly expressed in the cells of mineralized tissue. Clinically, ALP expression is used as a marker for GCNIS and seminoma. ALP activity was markedly higher in GCNIS tubules compared with normal seminiferous tubules as determined by NBT/BCIP staining (*Figure 6A*). All four types of ALPs were expressed in normal testis and the expression in GCNIS with microlithiasis and seminoma was not significantly higher, although some tumor specimens have marked expression of all four ALP isozymes. This shows that ALP activity is higher in GCNIS/TGCTs and typically mediated by a combination of different isozymes rather than a single type of ALP (*Figure 6B*). To determine if mineralization can be induced in germ cells, the mouse spermatogonial cell line GC1 was treated with increasing concentrations of calcium and phosphate, and mineral deposition was subsequently visualized by alizarin red. Mineralization was already observed after co-treatment with calcium and phosphate for 2 days, but also after 7 days of treatment with calcium or phosphate alone (*Figure 6C*). With a low serum phosphate concentration (0.9 mM), 3 mM calcium was required to obtain rapid mineralization after 2 days, whereas even 4 mM phosphate did not induce mineralization in a normal calcium concentration (1.8 mM) after 2 or 4 days. However, an increment in calcium to 2.0 mM enabled five of the six phosphate doses to induce rapid mineralization of the GC1 cells. This shows that low calcium levels are protective in the high intratesticular phosphate environment. Longer exposure (4 and 7 days) augmented the response in a dose-dependent manner. The addition of PPi, an inhibitor of mineralization, reduced mineralization, whereas co-treatment with PPi and pyrophosphatase (PPA1) that catalyzes the hydrolysis of PPi to inorganic phosphate (Pi) reintroduced mineralization (*Figure 6D*). To assess the regulation of phosphatase activity in germ cells, a human cell line TCam2 was used as a positive control. TCam2 is derived from a seminoma and retains high expression of ectopic ALP, as assessed by NBT/BCIP staining, which was critically dependent on an alkaline environment with marked phosphatase activity at pH 9, but not at pH 7 (*Figure 6E*). Staining with Fast Blue RR revealed that ALP activity was increased in the presence of osteogenic medium compared with control medium, indicating that germ cells increase their ALP activity in the presence of factors that facilitate bone differentiation (*Figure 6F*). However, an ex vivo model using wild-type mouse testicles (*Figure 7A*) cultured for 14 days with osteogenic medium -/+ phosphate did not change the expression of the bone markers *Bglap*, *Runx2*, or *Alpl* (*Figure 7B*). However, occasionally one or few tubules had expression of RUNX2, which was observed more often in specimens treated with osteogenic medium than in specimens treated with vehicle (*Figure 7C*). Few of the specimens only had Sertoli cells in the tubules due to loss of germ cells and were therefore excluded from the analysis. RUNX2 protein expression was not observed in those tissue pieces.

We propose that the formation of testicular microcalcifications occurs because of one or more of the following events: disturbed local phosphate homeostasis, decreased function of the Sertoli cells, reduction of mineralization inhibitors, or aberrant germ cell function. These changes alone or in combination can drive osteogenic-like differentiation of germ cells resulting in intratubular deposition of microcalcifications (*Figure 8*).

## Discussion

This translational study demonstrates that testicular microcalcifications, both benign and malignant, occur secondary to changes in gonadal phosphate homeostasis and are often accompanied by a subsequent osteogenic-like differentiation of germ cells or testicular somatic cells. Microcalcifications are more prevalent in testicular biopsies with GCNIS than without (*Kang et al., 1994*), which highlights that GCNIS or TGCTs release factors involved in the development of microlithiasis. Still, the accompanied impaired Sertoli cell function appears to be of greater importance than the presence of malignant germ cells. In this manuscript, we show that impaired Sertoli cell function in both mice and humans increases the risk of testicular microlithiasis and this association seems to be more persistent than the malignant linkage.

The systemic master regulator of phosphate, FGF23, is highly expressed in GCNIS, EC, and hESC but not in classical seminoma. This suggests that the expression of FGF23 is linked with the transformation of GCNIS to the invasive EC and is not a result of duplication of chromosome 12p which is also prevalent in classical seminoma. The marked expression in EC and hESCs highlights that FGF23

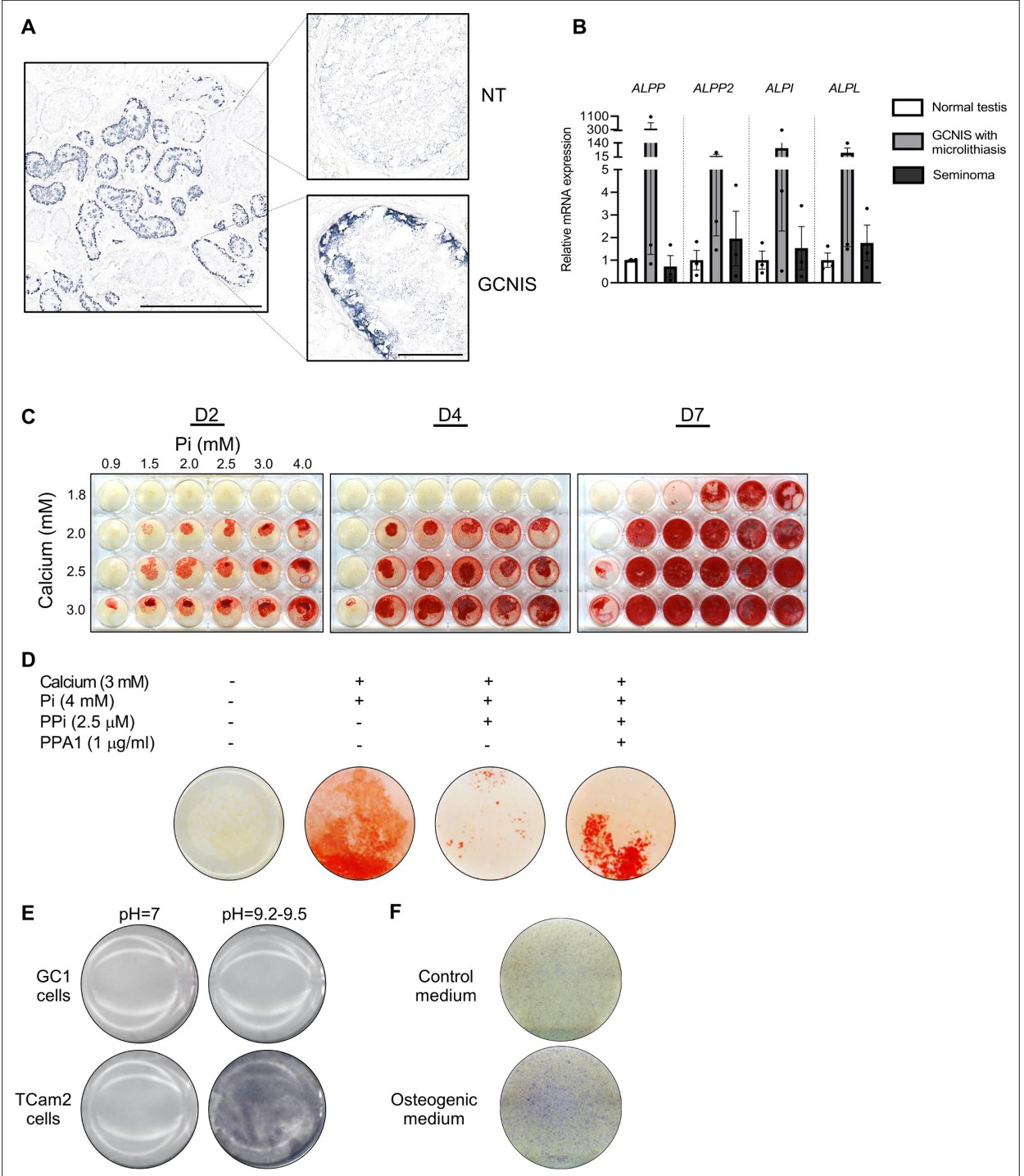

**Figure 6.** Alkaline phosphatase activity in normal testis and germ cell neoplasia in situ (GCNIS) and calcium and phosphate induced mineralization in vitro. (**A**) Staining of alkaline phosphatase activity with BCIP/NBT in normal testis (NT) with adjacent tubules containing GCNIS. Scale bars correspond to 1 mm (left picture) and 100 µm (right pictures). (**B**) Expression levels of the four alkaline phosphatases in normal testis, GCNIS with microlithiasis, and seminoma. ANOVAs with Dunnett's multiple comparisons test were used as the statistical tests. (**C**) Mineralization stained with alizarin red in GC1 cells treated with increasing concentrations of calcium and inorganic phosphate (Pi) for 2, 4, or 7 days. (**D**) Mineralization stained with alizarin red in GC1 cells with calcium and/or phosphate in the presence of the mineralization inhibitor pyrophosphate (PPi) and/or pyrophosphatase (PPA1) that catalyzes the hydrolysis of PPi to Pi for 4 days. (**E**) Staining of alkaline phosphatase activity with BCIP/NBT in GC1 or TCam2 cells in pH=7 or pH=9.2–9.5. (**F**) Fast Blue RR staining of alkaline phosphatase activity in GC1 cells cultured in control or osteogenic medium for 14 days.

The online version of this article includes the following source data for figure 6:

**Source data 1.** Numerical data used to generate the figure.

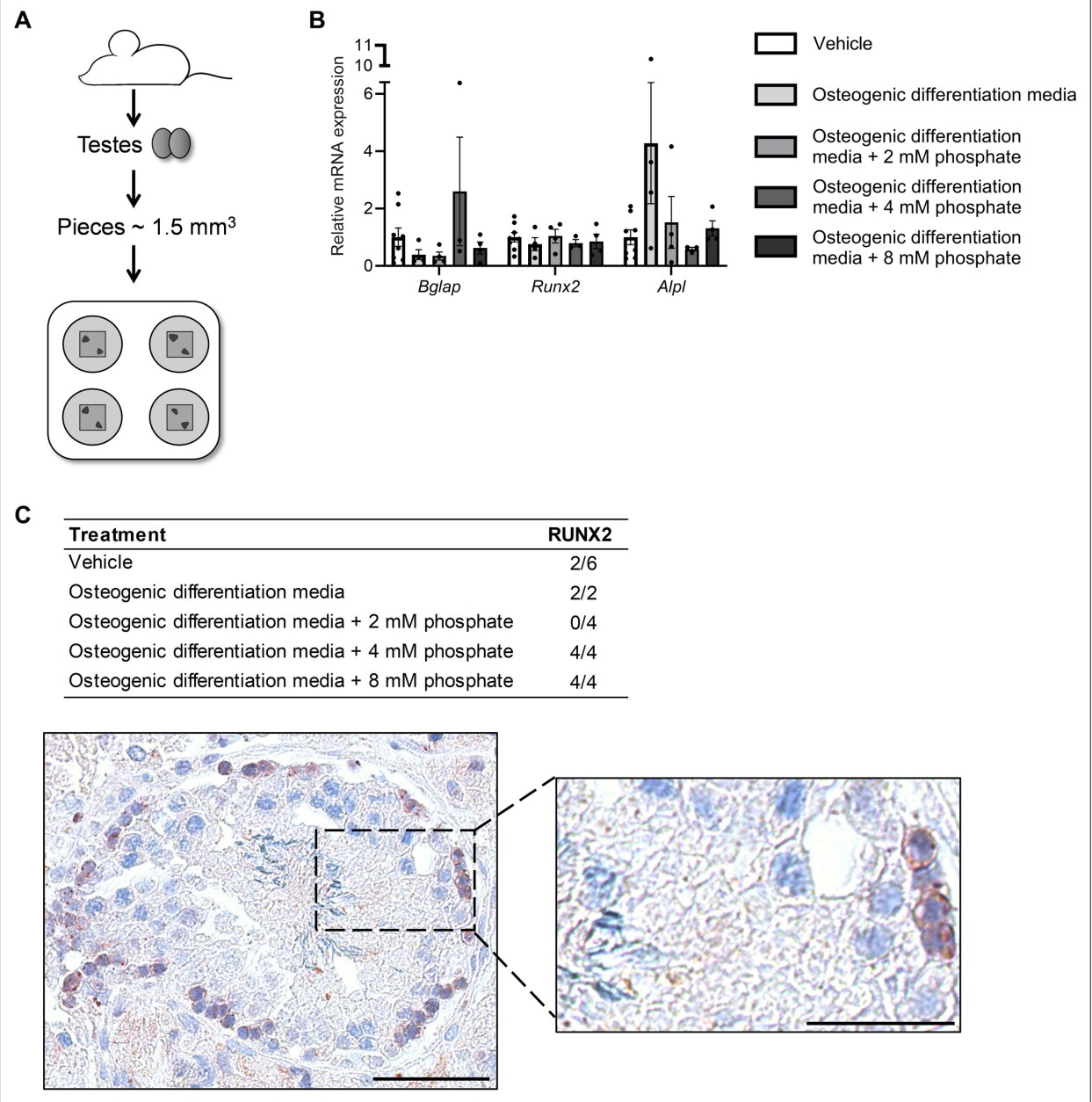

**Figure 7.** Ex vivo mouse testis culture in osteogenic differentiation media. (**A**) Schematic representation of the mouse testis ex vivo model. (**B**) Expression levels of *Bgalp*, *Runx2*, and *Alpl* in ex vivo mouse testis tissue culture specimens treated with osteogenic medium -/+ phosphate. ANOVAs on log-transformed (*Bglap* and *Alpl*) or non-transformed data were used as the statistical tests. (**C**) Table of the prevalence of focal expression of RUNX2 in mouse testis ex vivo cultures with vehicle and osteogenic treatment for 14 days. Representative RUNX2 staining in the ex vivo culture treated with osteogenic media and 8 mM phosphate. Scale bars correspond to 50 μm (left picture) and 25 μm (right picture).

The online version of this article includes the following source data for figure 7:

**Source data 1.** Numerical data used to generate the figure.

is an early embryonic signal, which is further supported by the strong correlation between *FGF23* and pluripotency factor *NANOG* in EC and the high *FGF23* expression in EC with polyembryoma formation. Moreover, it is in accordance with previous reports identifying *FGF23* transcripts in hESCs and during early embryonic life (***Cormier et al., 2005***; ***Wei et al., 2005***).

Phosphate levels in the testis are threefold higher than in serum (***Jenkins et al., 1980***; ***Jenkins et al., 1983***). In vascular and soft tissue calcification, FGF23-Klotho activity has been suggested to regulate ion metabolism and subsequently induce the presence of bone-like cells in the vessels

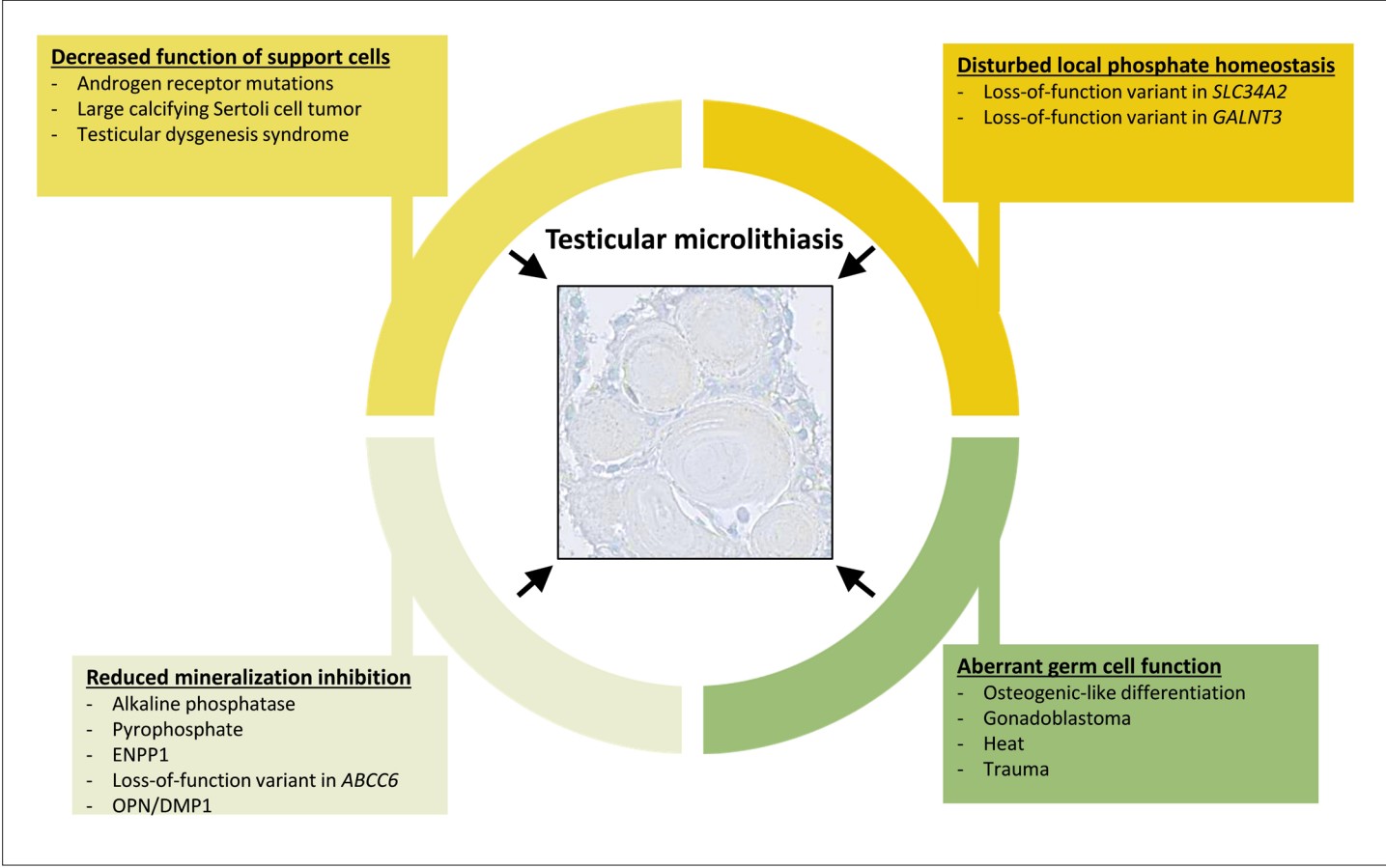

**Figure 8.** Overview of events that can contribute to the formation of testicular microlithiasis. We suggest that the formation of testicular microcalcifications occurs as a consequence of one or more of the following events: decreased function of the Sertoli cells, disturbed local phosphate homeostasis, reduction of mineralization inhibitors, or aberrant germ cell function. These changes alone or in combination can drive osteogenic-like differentiation of germ cells resulting in intratubular deposition of microcalcifications consisting of hydroxyapatite.

(**Memon et al., 2008**; **Leszczynska and Murphy, 2018**). The observed hydroxyapatite in the cauda epididymis of *Fgf23*$^{-/-}$ mice shows that loss of FGF23 signaling causes deposition of hydroxyapatite. As EC and GCNIS have no GalNAc-T3 expression (**Rajpert-De Meyts et al., 2007**), FGF23 will rapidly be cleaved into cFGF23 which is in accordance with the detection of mainly cFGF23 (and not iFGF23) in seminal fluid from GCNIS and EC patients. High intratesticular cFGF23 may bind the Klotho/FGFR1 receptor and antagonize the effect of iFGF23 (**Goetz et al., 2010**) thereby inducing a phenotype resembling *Fgf23*$^{-/-}$ mice with increased expression of *Bglap*. However, in our human testis ex vivo model neither cFGF23 nor iFGF23 induced changes in phosphate transporters or *BGLAP* suggesting that short-term exposure (24 hr) to high cFGF23 does not induce changes in phosphate transporters, bone factors, or microcalcifications. The epididymal phenotype of *Fgf23*$^{-/-}$ mice does not appear to be mediated via systemic hyperphosphatemia as a high phosphate dietary exposure for weeks did not result in testicular or epididymal microcalcifications or changes in testicular phosphate transporter expression. Instead, we have previously shown that germ cell-specific deletion of Klotho in mice leads to aberrant mineral homeostasis, particularly calcium transport through transient receptor potential cation channel subfamily V 5 (TRPV5) (**Bøllehuus Hansen et al., 2020**), which may be important as low calcium appears to be protective for inducing microcalcifications in a high phosphate environment. Local testicular mineral levels depend mainly on the presence and activity of specific transporters and sensors of calcium and phosphate in the reproductive organs, rather than on systemic concentrations (**Boisen et al., 2021**). This also explain why patients with loss-of-function variants in the predominant testicular phosphate transporter *SLC34A2* present with testicular microcalcifications (**Corut et al., 2006**). Moreover, patients with loss-of-function variants in *GALNT3*, which lead to the

early degradation of iFGF23 to its C-terminal form, present with global calcifications, severe testicular microlithiasis, and microcalcifications (*Garringer et al., 2007*; *Campagnoli et al., 2006*).

*Fgf23*-/- mice exhibit hypogonadism (*DeLuca et al., 2008*) and spermatogenic arrest, which resembles the condition of some men with testicular dysgenesis syndrome that occasionally have testicular microcalcifications despite no malignancy (*Pedersen et al., 2016*; *Rebourcet et al., 2014a*). Benign microcalcifications may be facilitated by low levels of androgens or gonadotropins as such hormonal imbalances can disrupt the Sertoli cell function. As a result, the germ cells may remain in their prepubertal stem cell state, rather than maturing properly leaving them susceptible to external stimuli that could trigger abnormal differentiation. Changes in the local testicular phosphate concentration (which is threefold higher than serum) could exert such a stimuli (*Jenkins et al., 1980*; *Jenkins et al., 1983*; *Sharpe et al., 2003*; *O'Shaughnessy et al., 2010b*). This hypothesis was supported by the presence of microcalcifications in *hpg* mice with concomitant global *AR* ablation as these mice have immature Sertoli cells and are unable to complete spermatogenesis (*O'Shaughnessy et al., 2009*). Moreover, these mice had altered testicular phosphate homeostasis characterized by a nonsignificant tendency toward lower *Slc34a2*, but significantly higher *Slc34a1* expression compared with WT mice. However, both *hpg* and *hpg* mice with Sertoli cell-specific *AR* deletion (*hpg.SCARKO*) had similar changes in phosphate transporters and only 30–36% frequency of microlithiasis, suggesting that additional mechanisms are contributing to the formation of microlithiasis. The bone marker *Bglap* was detected exclusively in all the *hpg* mice with concomitant global *AR* deletion (*hpg.ARKO*) but not in *hpg* mice with intact AR, which supports that some testicular cells undergo osteogenic trans-differentiation, and this is related to more abundant deposition of hydroxyapatite. We cannot determine the cell-of-origin, but we have previously shown that vitamin D induces an osteogenic-like differentiation of NTera2 cells which start to express bone-specific proteins such as Osteocalcin (*Blomberg Jensen et al., 2012*). However, the osteogenic-like differentiation in *hpg* mice occurs in non-malignant prepubertal/non-maturated testicular cells.

Peritubular cells could be a good candidate because they are mesenchymal-derived and such fibroblast-like cells can undergo osteogenic-like differentiation during vascular calcification at other sites of extra-skeletal mineralization (*Ronchetti et al., 2013*). However, human Sertoli cells can also form large-cell calcifying tumors (*Gourgari et al., 2012*), indicating that multiple gonadal cell types have the potential to become bone-like. In mice, Sertoli cell ablation before puberty caused extensive intratubular microcalcifications (*Rebourcet et al., 2014b*), whereas Sertoli cell ablation in adulthood did not result in the formation of microcalcifications when assessed 3 months later despite a greater burden of immediate intratubular cell death. This finding is important because it demonstrates that microcalcifications are not caused by massive intratubular cell death in mice but are critically dependent on germ cells with stem cell potential. Sertoli cell maturation and function seem essential for protection against osteogenic-like differentiation of the germ cells or peritubular cells and benign microcalcifications are more likely to occur due to prepubertal Sertoli cell disturbance. The spermatogonial stem cells may also be the cell type that undergoes osteogenic-like differentiation as they can spontaneously develop into all three germ layers when isolated from adult mice, form teratomas in nude mice, and participate in the formation of multiple organs when injected into blastocytes (*Guan et al., 2006*). RUNX2 is an essential transcription factor for osteoblast development, and although germ cells express a different isoform of RUNX2 (lacking exons 4–8), it is transcribed from the same promoter as in bone (*Ogawa et al., 2000*; *Jeong et al., 2008*). RUNX2 was undetectable in healthy seminiferous tubules using an antibody targeting exon 5, but was detected in GCNIS, hyalinized/calcified tissues, and the otherwise seemingly healthy seminiferous tubules adjacent to bone in the testis, indicating that the bone-specific RUNX2 isoform is also expressed in the testis when microcalcifications occur. Also, the mature bone marker Osteocalcin was expressed in a fraction of cells that morphologically resemble germ cells. Unfortunately, there was no cell-tracing marker in the mouse models so the required tracing experiments to support that these bone-like cells originated from the germ cells could not be conducted. Instead, a spermatogonial mouse cell line was exposed to calcium and phosphate at concentrations like those found in the testis or the epididymis. After 2–7 days of exposure, the spermatogonia started to deposit minerals illustrating that they developed some osteoblast-like characteristics, but they did not undergo complete osteogenic differentiation. A similar experiment using an adult mouse testis culture model without exposure to gonadotropins or androgens showed that treatment with an osteogenic cocktail for 14 days did not induce *Runx2*, *Bgalp*, or *Alpl* but a focal

expression of the RUNX2 (using the antibody targeting exon 5) was found in specimens treated with both vehicle and the osteogenic cocktail. This suggests that RUNX2 expression was switched to the bone-specific isoform (*Kanto et al., 2016*) during ex vivo culturing and in accordance RUNX2 was more frequently found in specimens treated with the osteogenic cocktail+high phosphate.

One of the most potent mineralization inhibitors is PPi, which can be degraded by ALP or pyrophosphatase (*Orriss, 2020*). PPi prevented the mineralization induced by calcium and phosphate in the GC1 cell line, which was reintroduced in the presence of pyrophosphatase-degrading PPi. This highlights the role of PPi as an inhibitor of spermatogonial mineralization. ALPs convert PPi to Pi and are expressed by primordial and malignant germ cells (*Hofmann et al., 1989*; *Jacobsen and Nørgaard-pedersen, 1984*). Four genes encode ALPs in humans (five in mice), and all are expressed in malignant germ cells, including the tissue nonspecific alkaline phosphatase (TNAP) (*Hofmann et al., 1989*). TNAP is required for normal bone formation and regulates vascular matrix mineralization by inactivation of the mineralization inhibitors PPi and OPN (*Lomashvili et al., 2004*). We show here that TCam2 cells have high ALP activity in alkaline pH, suggesting that mineralization is promoted by malignant cells. This implies that men with GCNIS and TGCTs are prone to microcalcifications due to lower PPi levels caused by increased ALP activity. Several other inhibitors and promoters of mineralization could be involved, but the marked epididymal mineralization observed in $Fgf23^{-/-}$ mice, despite the presence of SIBLING proteins that inhibit hydroxyapatite formation and the extensive testicular microcalcifications in men with loss-of-function variants in *SLC34A2* (*Corut et al., 2006*), suggesting that aberrant function of just one component can be enough to promote the formation of microcalcifications. Immature bone formation in the testis is commonly observed in teratomas but can occur without malignancy. Mature bone tissue formation has also been reported in the testis after trauma or bleeding without signs of GCNIS or invasive TGCTs (*Yoneda et al., 1979*). Here, we also show intratesticular bone in a specimen despite no detection of GCNIS or malignant cells. A comprehensive analysis of germ cells adjacent to the bone tissue revealed focally bone-specific RUNX2 positive germ cells but none of them expressed the other investigated bone markers or deposition of hydroxyapatite, indicating that the mature bone has been formed in the interstitial compartment through a different mechanism related to the trauma rather than alterations in mineral homeostasis leading to testicular microcalcifications.

In conclusion, we show that the formation of testicular microcalcifications occurs due to either malignant or benign etiology, and includes one or more of the following events: aberrant function of Sertoli cells, disturbed local phosphate homeostasis, change in mineralization inhibitors, or aberrant germ cell function. These changes alone or in combination can drive osteogenic-like differentiation of germ cells resulting in intratubular deposition of microcalcifications consisting of hydroxyapatite. Thus, this study suggests that microcalcifications alone should not be considered a marker for malignancy in the testis.

## Materials and methods
### Human tissue samples
Patients were recruited from the Department of Growth and Reproduction, Rigshospitalet, Denmark, in accordance with the Helsinki Declaration after approval from the local ethics committee (Permit No. H-15000931, H-17004362, KF 01 2006-3472). Adult testis samples were obtained from orchiectomy specimens performed due to TGCTs. The tissue surrounding the tumor contained tubules with either GCNIS cells or normal/impaired spermatogenesis. Each sample was divided into fragments, which were either snap-frozen and stored at –80°C for RNA extraction or fixed overnight at 4°C in formalin or paraformaldehyde and were subsequently embedded in paraffin. An experienced pathologist evaluated all samples and IHC markers were used to examine the histological subtypes of the TGCT. Fetal gonads were collected after elective terminations for nonmedical reasons and gestational age was estimated by ultrasound at the Department of Gynecology (Rigshospitalet/Hvidovre Hospital, Denmark) after approval by the local ethics committee (H-1-2012-007).

### Microarray
For this study, the data were extracted from microarray analysis, which was performed and published in full previously (*Sonne et al., 2009*). Briefly, snap-frozen tissues were fixed in 75% ethanol and analyzed

for AP-2γ (#sc-12762, Santa Cruz, RRID:AB_667770) by IHC to identify gonocytes and GCNIS and a slide was stained for ALP activity detectable in GCNIS and fetal germ cells. Before microdissection, slides were stained with nitroblue tetrazolium (NBT)-5-bromo-4-chloro-3-indolyl phosphate (BCIP). RNA was purified using Ambion RNAqueous micro kit (Applied Biosystem/Ambion). The quality of RNA was evaluated using Bioanalyzer Picokit (Agilent Technologies). RNA samples were amplified in two rounds using the MessageAMP II Kit (Applied Biosystem/Ambion). RNA was reverse-transcribed using 50 ng/µl random hexamer primers. Analyses were conducted on three hESCs, microdissected fetal gonocytes, microdissected GCNIS, and tissue containing different percentages of GCNIS and normal testis samples. Agilent Whole Human Genome Microarray 4×44 K chips, Design Number 014850 (Agilent Technologies), were used for arrays. Hybridization and scanning of one-color arrays were performed as described by the manufacturer and analyzed using Agilent Feature Extraction software (version 9.1.3.1, RRID:SCR_014963).

## Quantitative RT-PCR

RNA was extracted from 13 tissue specimens with classical seminoma, 10 ECs where some included components of teratoma, and 30 GCNIS samples (GCNIS adjacent to EC, seminoma, mixed tumor, and six GCNIS samples without an overt tumor). RNA from normal testis was purified from three orchiectomy specimens with seminiferous tubules containing varying degrees of full and partial spermatogenesis but with no GCNIS or tumor. Two normal testis RNA samples were purchased from different companies (Applied Biosystems/Ambion and Clontech). RNA and cDNA preparation and quantitative RT-PCR (qRT-PCR) were conducted as described previously (*Blomberg Jensen et al., 2010a*). qRT-PCR was performed using primers listed in *Supplementary file 1*. The representative bands from each primer combination were sequenced for verification (Eurofins MWG GmbH). qRT-PCR analyses were performed twice in duplicates on two different plates using a Stratagene Mx300P cycler with SYBR Green QPCR Master Mix (Stratagene). Changes in gene expression were determined with the comparative CT method using glyceraldehyde-3-phosphate dehydrogenase (*GAPDH*) (ALP expression in human normal testis, GCNIS with microlithiasis, and seminoma) or *β2-microglobulin* (*β2M*) (all other gene expression analyses) as control genes. One sample in the *hpg*.ARKO group was lost during analyses, and *Slc34a1* was therefore not determined in that mouse.

## Western blot

Frozen tissues prepared for western blot included adult normal testis, testis with GCNIS, homogeneous anaplastic seminoma, EC samples, and cell lysates from NTera2 and TCam2 cells. The tissues were homogenized in RIPA lysis buffer with protease inhibitor cocktail set lll (Calbiochem). After centrifugation, supernatants were collected and diluted in SDS Sample buffer and heated for 5 min at 95°C. Proteins were separated on an 8–16% gradient gel (Pierce) and transferred to a PVDF membrane (Bio-Rad). The membrane was blocked in 5% milk for 1 hr and incubated overnight with primary antibody (diluted 1:200, #sc-16849, RRID:AB_2104631). β-Actin was used as loading control (diluted 1:200, #sc-47778, RRID:AB_626632). The membranes were further incubated for 1 hr with an HRP-conjugated secondary antibody. Membranes were washed in TBST, and protein signals were detected using one-step NBT/BCIP (Pierce).

## Immunohistochemistry

IHC staining was performed as described previously (*Blomberg Jensen et al., 2010b*). Briefly, antigen retrieval was conducted by microwaving for 15 min in TEG or CIT buffer before incubation with 2% non-immune goat serum (Zymed Histostain kit) or 0.5% milk powder diluted in TBS. Subsequently, slides were incubated with primary antibodies overnight at 4°C (*Supplementary file 2*) followed by a secondary antibody, and a peroxidase-conjugated streptavidin complex (Zymed Histostain kit) was used as a tertiary layer. Visualization was performed with amino ethyl carbazole. Between incubation steps, the slides were washed with TBS. Counterstaining was performed with hematoxylin eosin (HE) staining. All experiments were performed with control staining without primary antibody. Staining was classified according to an arbitrary semiquantitative reference scale depending on the intensity of staining and the proportion of cells stained: +++, strong staining in nearly all cells; ++, moderate staining in most of the cells; +, weak staining or a low percentage of cells stained; +/-, very weak

staining in single cells; none, no positive cells detected. All antibodies used have been validated by testing in positive control tissue and/or western blot or in situ hybridization.

## NTera2 xenografts and treatments

All animal experiments were performed in compliance with the Danish Animal Experiments Inspectorate (license number 2011/561-1956) and conducted as previously described (*Blomberg Jensen et al., 2012*). Mice were housed and interventions were conducted at Pipeline Biotech in a pathogen-free area. NTera2 cells (RRID:CVCL_3407) were grown under standard conditions in 175 cm$^2$ flasks and media were changed every 48 hr. Matrigel (BD Biosciences) was diluted in Dulbecco's modified Eagle's medium (DMEM) before being mixed 1:1 with NTera2 cells. NTera2 cells ($5×10^6$ cells) were injected subcutaneously into the flanks of male nude mice 6- to 8-week-old NMRI mice (Fox1$^{nu}$, Taconic Europe, RRID:IMSR_TAC:NMRINU). The NTera2 cells were grown on the mice for 28–56 days. Tumor volumes were calculated from two tumor diameter measurements using a Vernier caliper: tumor volume = L × W × ½W. If a tumor diameter of 12 mm was reached the animals were sacrificed. At the end of the study, mice were euthanized according to the animal welfare euthanasia statement. Tumors, kidneys, and testes were harvested and weighed, and one half was fixed in formalin and embedded in paraffin for IHC, and the other half was snap-frozen.

## Hypogonadal mice with androgen receptor ablation globally or specifically in the Sertoli cells and Sertoli cell-ablated mice

The *hpg* mice were bred in Glasgow and all procedures were carried out under UK Home Office License and with the approval of a local ethical review committee. *hpg*.SCARKO and *hpg*.ARKO mice were generated as previously described (*O'Shaughnessy et al., 2009*; *O'Shaughnessy et al., 2010a*). SCARKO and ARKO mice were generated by crossing mice carrying floxed *Ar* (*Ar*$^{fl}$) with mice expressing Cre regulated by the Sertoli-specific promoter of AMH (*Amh-Cre*) or the ubiquitous promoter PGK-1. *hpg*.SCARKO mice were generated by crossing female *Ar*$^{fl/fl}$ mice heterozygous for the GnRH deletion (*hpg*/+) (C3HE/HeH-101/H) with male *hpg*/+mice expressing *Amh-Cre*. The *hpg*.ARKO was generated the same way using a *Pgk1-Cre* instead of *Amh-Cre*. All mice were male and euthanized at 9–11 weeks of age by cervical dislocation. Sertoli cell-ablated mice were generated as previously described (*Rebourcet et al., 2014b*). Briefly, *Amh-Cre* mice were bred with diphtheria-toxin receptor (iDTR) mice, and the offspring were injected subcutaneously with a single acute dose (100 ng in 50 µl) of diphtheria-toxin or with 50 µl sterile water (vehicle) at postnatal day 18. The mice were euthanized on postnatal day 80.

## *Fgf23*$^{-/-}$ mice

*Fgf23*$^{-/-}$ mice were generated as previously described (*Sitara et al., 2004*). Mice were bred in Boston and all studies were approved by the Institutional Animal Care and Use Committee of Harvard. The mice were male and euthanized at 8 weeks of age by using carbon dioxide followed by cervical dislocation.

## Biochemical analyses

Human iFGF23 and total FGF23 (iFGF23+cFGF23) were measured in duplicates in batched assays using assays from Immutopics (Immutopics, #60-6100, #60-6600) according to the manufacturer's instructions. The seminal fluid FGF23 measurements in healthy men have been published previously (*Bøllehuus Hansen et al., 2020*). Total serum calcium and phosphorus levels were determined using Stanbio LiquiColor Kits (Stanbio Laboratory).

## In vitro cultures

hESC lines were cultured in Sheffield as described previously (*Andrews et al., 1982*). H7 cells with (karyotype of 47,XX,+del(1)(p22p22),der(6)t(6;17)(q27;q1)) or without (karyotype of 46 XX) genomic aberrations were sorted using a fluorescence-activated cell sorter and only cells expressing the pluripotency marker SSEA3 (undifferentiated cells) were analyzed in addition to H7 pool of unsorted cells. GC1 cells (ATCC CRL-2053, RRID:CVCL_8872) were grown under standard conditions at 37°C at 5% $CO_2$ in DMEM supplemented with penicillin (100 U/ml) and streptomycin (100 mg/ml) (Gibco), 10% FBS (Gibco, #11573397), and L-glutamine (Gibco, #25030–024) (2 mM final conc.) was also added

to the media. To investigate calcification in the GC1 cells, the cells were grown in 24-well plates and treated with increasing concentrations of $CaCl_2$ and $NaH_2PO_4 \cdot Na_2HPO_4$ alone or in combination with PPi (Merck, #P8010) and/or phosphatase (Prospec, #ENZ-241). The medium was changed every second day. Human and mouse testes were cultured ex vivo using a hanging drop culture approach (*Jørgensen et al., 2014*) or culture on agarose gel pieces, respectively. In human testis specimens, the effects of 50 ng/ml iFGF23 or 200 ng/ml cFGF23 were investigated after being added to culture media with 0.1% BSA for 24 hr. In the mouse ex vivo cultures, testis from adult 8-week-old mice were dissected and treated with osteogenic medium in a gel-based tissue model. Briefly, agarose gels (1.5%) were cut into $8 \times 8 \times 8$ mm$^3$ cube, placed in four-well plates, and soaked in culture media (DMEM-F12, 1× penicillin/streptomycin, 1× insulin, transferrin and selenium supplement [ITSS], 10% fetal bovine serum [FBS]) for a minimum of 24 hr before setup of tissue cultures. Mouse testis was dissected into 1.5 mm$^3$ pieces and placed in opposite corners on each gel. The tissues were cultured for 14 days at 34°C and 5% $CO_2$ in 325 µl media with osteogenic medium containing 50 µg/ml ascorbic acid (Merck, #A4544), 10 mM β-glycerol phosphate (Merck, #G9422), and either -/+2, 4, or 8 mM phosphate or vehicle. The medium was changed every second day. At least three replicate tissue pieces from the same mouse and treatment group were fixed in formalin followed by paraffin embedding for IHC and histological analyses. Tissue was evaluated for morphology and technical quality based on HE staining. TCam2 cells (RRID:CVCL_T012) were grown under standard conditions in 175 cm$^2$ flasks and media were changed every 48 hr.

## von Kossa, alizarin red, BCIP/NBT, and Fast Blue RR staining

For von Kossa staining of tissue sections, the paraffin sections were first deparaffinized and rinsed in ddH$_2$O. Sections were then incubated with 1% silver nitrate solution under ultraviolet light for 20 min. Thereafter, the sections were rinsed in ddH$_2$O and incubated with 5% sodium thiosulfate for 5 min and rinsed in ddH$_2$O (and sometimes counterstained with Mayer's hematoxylin) and dehydrated through graded alcohol and cleared in xylene. For alizarin red staining of tissue sections, deparaffinized sections were stained with 2% alizarin red solution (pH 4.1–4.3) for 5 min. The sections were then dehydrated in acetone and acetone-xylene (1:1) and cleared in xylene. For alizarin red staining of GC1 cells, the cells were washed in PBS and fixed in 10% formalin for 10 min. Thereafter, the cells were washed in ddH$_2$O and incubated with 0.5% alizarin red (pH 4.1–4.3) for 20 min protected from light with light shaking. Cells were washed two to three times with ddH$_2$O and left to dry. Cryosections (10 µm) were treated with BCIP/NBT mixture for 90 s. The BCIP/NBT mixture consisted of 45 µl BCIP stock solution and 35 µl NBT stock solutions in 10 ml water. Stock solutions: 50 mg/ml BCIP (Sigma #B8503) in 100% dimethylformamide (Sigma #D4254) and 75 mg/ml NBT (Sigma #N6876) in 70% vol/vol dimethylformamide/distilled water. The reaction was stopped in water (1 min). BCIP/NBT staining of cell lines was performed the same way but in citrate buffer (pH = 7) or revelation buffer (pH = 9.2–9.5). For Fast Blue RR staining, the cells were washed in PBS and then fixed with 3.7% formalin for 10 min. Then, the cells were rinsed in PBS and stained for 15 min at RT lightly shaking and protected from light. The cells were stained for ALP activity with Napthol AS-MX, 144 *n,n*-dimethylformamide and Fast Blue RR salt (Sigma #F0500).

## Statistics

Data were analyzed using GraphPad Prism v. 8 (GraphPad Software). Evaluation of Gaussian distribution was performed and if the residuals did not have Gaussian distribution or unequal variances, variables were transformed with natural logarithm and reevaluated. Hence, a two-sided unpaired Student's t-test was used between two groups of mice except for serum calcium between WT and *Fgf23*$^{-/-}$ where a Mann-Whitney test was used. One-way ANOVAs followed by Dunnett's multiple comparisons test were used to test for differences between gene expression in human normal testis and tumors, ex vivo cultures of human and mouse testes, and between WT and the different *hpg* mouse models. Kruskal-Wallis test with Dunn's multiple comparisons test was used to test for differences between levels of iFGF23 and total FGF23 (iFGF23+cFGF23) in seminal fluid from healthy men and tumor patients. For correlation analyses, residual plots were evaluated to ensure the validity of the correlation analyses. As a result, the Pearson correlation test was used to investigate the correlation between *FGF23* and the pluripotency genes *POU5F1* or *NANOG* in EC. Statistical significance was determined at the following levels: not significant p>0.05, *p<0.05, **p<0.01, ***p<0.001, and

****p<0.0001. Sample-size estimation was based on similar type of experimental design conducted previously by us or others.

## Acknowledgements

We thank Giulio Spagnoli for the MAGE antibody. We gratefully acknowledge several urologists and pathologists of the Greater Copenhagen area hospitals for their help with collecting the tissue samples. We thank Betina F Nielsen, Bonnie Håkansson, Ana R Nielsen, and Brian V Hansen for their skillful technical assistance. We appreciate the efforts of Carsten L Buus and Klaus Kristensen from Pipeline Biotech, who performed the xenografting. We also thank all the patients and donors who took part in this study. Danish Cancer Society, Forskningsrådet for sundhed og sygdom, Novo Nordisk Foundation, Aase og Ejnar Danielsens fond, Hørslev Fonden, Dagmar Wilhelms fond, and Ib Henriksens fond.

## Additional information

### Competing interests

Ewa Rajpert-De Meyts: Guest editor <i>eLife</i>. The other authors declare that no competing interests exist.

### Funding

| Funder | Grant reference number | Author |
|---|---|---|
| Danish Cancer Society Research Center | | Martin Blomberg Jensen |
| Independent Research Fund Denmark | | Martin Blomberg Jensen |
| Novo Nordisk Foundation | | Martin Blomberg Jensen |
| Aase and Ejnar Danielsens Fund | | Ida Marie Boisen |
| Hørslev Fonden | | Martin Blomberg Jensen |
| Dagmar Wilhelms Fund | | Martin Blomberg Jensen |
| Ib Henriksens Fund | | Martin Blomberg Jensen |

The funders had no role in study design, data collection and interpretation, or the decision to submit the work for publication.

### Author contributions

Ida Marie Boisen, Conceptualization, Data curation, Formal analysis, Funding acquisition, Validation, Investigation, Visualization, Methodology, Writing – original draft, Project administration, Writing – review and editing; Nadia Krarup Knudsen, John E Nielsen, Data curation, Methodology, Writing – review and editing; Ireen Kooij, Mathilde Louise Bagger, Jovanna Kaludjerovic, Noriko Ide, Birgitte G Toft, Arnela Mehmedbasic, Richard Norman, Data curation, Writing – review and editing; Peter O'Shaughnessy, Data curation, Supervision, Methodology, Writing – review and editing; Peter W Andrews, Lee B Smith, Data curation, Supervision, Methodology; Anders Juul, Ewa Rajpert-De Meyts, Supervision, Writing – review and editing; Anne Jørgensen, Data curation, Supervision, Writing – review and editing; Beate Lanske, Supervision, Methodology, Writing – review and editing; Martin Blomberg Jensen, Conceptualization, Data curation, Formal analysis, Supervision, Funding acquisition, Validation, Investigation, Methodology, Project administration, Writing – review and editing

### Author ORCIDs

Ida Marie Boisen http://orcid.org/0000-0003-2401-2877
Nadia Krarup Knudsen http://orcid.org/0009-0005-1591-343X
Peter O'Shaughnessy http://orcid.org/0000-0001-5651-7883
Arnela Mehmedbasic http://orcid.org/0009-0006-3680-247X

Martin Blomberg Jensen  https://orcid.org/0000-0003-3800-4253

### Ethics

Patients were recruited from the Department of Growth and Reproduction, Rigshospitalet, Denmark in accordance with the Helsinki Declaration after approval from the local ethics committee (Permit No. H-15000931, H-17004362, KF 01 2006-3472).

The hpg mice were bred in Glasgow and all procedures were carried out under UK Home Office License and with the approval of a local ethical review committee. Mice were bred in Boston and all studies were approved by the Institutional Animal Care and Use Committee of Harvard. All other animal experiments were performed in compliance with the Danish Animal Experiments Inspectorate (license number 2011/561-1956).

Reviewer #1 (Public review): https://doi.org/10.7554/eLife.95545.3.sa1
Author response https://doi.org/10.7554/eLife.95545.3.sa2

## Additional files

### Supplementary files

MDAR checklist

Supplementary file 1. List of primary antibodies used for IHC staining.

Supplementary file 2. List of primers used for quantitative RT-PCR (qRT-PCR).

### Data availability

All data generated or analysed during this study are included in the manuscript and supporting files; source data files have been provided for Figures 1-7.

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
