## [Editor Report · eLife Assessment]

This **valuable** study reports the link between a disruption in testicular mineral (phosphate) homeostasis, FGF23 expression, and Sertoli cell dysfunction. The data supporting the conclusion are **solid**. This work will be of interest to biomedical researchers working on testis biology and male infertility. The assessment is based on the editors' critical evaluation of the authors' responses.

---

## [Referee Report · Reviewer #1 (Public review)]

The authors have strengthened their conclusions by providing additional information about the specificity of their antibodies, but at the same time the authors have revealed concerning information about the source of their antibodies.

It appears that many of the antibodies used in this study have been discontinued because the supplier company was involved in a scandal of animal cruelty and all their goats and rabbits Ab products were sacrificed. The authors acknowledge that this is unfortunate but they also claim that the issue is out of their hands.

The authors' statement is false; the authors ought to not use these antibodies, just as the providing company chose to discontinue them, as

those antibodies are tied to animal cruelty. The issue that the authors feel OK with using them is of concern. In short, please remove any results from unethical antibodies.

Removal of such results also best serves science. That is, any of their results using the discontinued antibodies means that the authors' results are non-reproducible and we should be striving to publish good, reproducible science.

For the antibodies that do not have unethical origins the authors claim that their antibodies have been appropriately validated, by "testing in positive control tissue and/or Western blot or in situ hybridization". This is good but needs to be expanded upon. It is a strong selling point that the Abs are validated and I want to see additional information in their Supplementary Table 2 stating for each Ab specifically:

(1) What +ve control tissue was used in the validation of each Ab and which species that +ve control came from. Likewise, if competition assays to confirm validity was used, please also specify.

(2) Which assay was the Ab validated for (WB, IHC, ELISA, all etc)

(3) For Antibodies that were validated for, or using WBs please let the reader know if there were additional bands showing.

(4) Include references to the literature that supports these validations. That is, please make it easy for the reader to appreciate the hard work that went into the validation of the Antibodies.

Finally, for the Abs, when the authors write that "All antibodies used have been validated by testing in positive control tissue and/or Western blot or in situ hybridization" I fail to understand what in situ hybridisation means in this context. I am under the impression that in situ hybridisation is some nucleic acid -hybridising-to-organ or tissue slice. Not polypeptide binding.

---

## [Author Response]

The following is the authors’ response to the current reviews.

**Reviewer #1 (Public review):**
The authors have strengthened their conclusions by providing additional information about the specificity of their antibodies, but at the same time the authors have revealed concerning information about the source of their antibodies.It appears that many of the antibodies used in this study have been discontinued because the supplier company was involved in a scandal of animal cruelty and all their goats and rabbits Ab products were sacrificed. The authors acknowledge that this is unfortunate but they also claim that the issue is out of their hands.The authors' statement is false; the authors ought to not use these antibodies, just as the providing company chose to discontinue them, as those antibodies are tied to animal cruelty. The issue that the authors feel OK with using them is of concern. In short, please remove any results from unethical antibodies.Removal of such results also best serves science. That is, any of their results using the discontinued antibodies means that the authors' results are non-reproducible and we should be striving to publish good, reproducible science.For the antibodies that do not have unethical origins the authors claim that their antibodies have been appropriately validated, by "testing in positive control tissue and/or Western blot or in situ hybridization". This is good but needs to be expanded upon. It is a strong selling point that the Abs are validated and I want to see additional information in their Supplementary Table 2 stating for each Ab specifically:(1) What +ve control tissue was used in the validation of each Ab and which species that +ve control came from. Likewise, if competition assays to confirm validity was used, please also specify.(2) Which assay was the Ab validated for (WB, IHC, ELISA, all etc)(3) For Antibodies that were validated for, or using WBs please let the reader know if there were additional bands showing.(4) Include references to the literature that supports these validations. That is, please make it easy for the reader to appreciate the hard work that went into the validation of the Antibodies.Finally, for the Abs, when the authors write that "All antibodies used have been validated by testing in positive control tissue and/or Western blot or in situ hybridization" I fail to understand what in situ hybridisation means in this context. I am under the impression that in situ hybridisation is some nucleic acid -hybridising-to-organ or tissue slice. Not polypeptide binding.**Recommendations for the authors**:
**Reviewer #1 (Recommendations for the authors):**
Remove results that have been obtained by unethically-sourced antibody reagents.Strengthen the readers' confidence about the appropriateness & validity of your antibodies.

First, we want to stress that reviewer 1 has raised his critique related to the used of antibodies from Santa Cruz biotechnology not only through the journal. The head of our department and two others were contacted by reviewer 1 directly without going through the journal or informing/approaching the corresponding or first author. It is our opinion that this debate and critique should be handled through the journal and editorial office and not with people without actual involvement in the project.

It is correct that we have purchased antibodies from Santa Cruz Biotechnologies both mouse, rabbit and goat antibodies as stated in the correspondence with the reviewer.

As stated in our previous rebuttal – the goat antibodies from Santa Cruz were discontinued due to inadequate treatment of goats after settling with the authorities in 2016.

https://www.nature.com/articles/nature.2016.19411

https://www.science.org/content/blog-post/trouble-santa-cruz-biotechnology

We have used 11 mouse, rabbit or goat antibodies from Santa Cruz biotechnologies in the manuscript as listed in supplementary table 2 of the manuscript and all of them have been carefully validated in other control tissues supported by ISH and/or WB and many of them already used in several publications by our group https://pubmed.ncbi.nlm.nih.gov/34612843/, https://pubmed.ncbi.nlm.nih.gov/33893301/, https://pubmed.ncbi.nlm.nih.gov/32931047/, https://pubmed.ncbi.nlm.nih.gov/32729975/, https://pubmed.ncbi.nlm.nih.gov/30965119/, https://pubmed.ncbi.nlm.nih.gov/29029242/, https://pubmed.ncbi.nlm.nih.gov/23850520/, https://pubmed.ncbi.nlm.nih.gov/23097629/, https://pubmed.ncbi.nlm.nih.gov/22404291/, https://pubmed.ncbi.nlm.nih.gov/20362668/, https://pubmed.ncbi.nlm.nih.gov/20172873/, and other research groups. All antibodies used in this manuscript were purchased before the whole world was aware of mistreatment of goats that was evident several years later.

We do not support animal cruelty in anyway but the purchase of antibodies from Santa Cruz biotechnologies were conducted long before mistreatment was reported. Moreover, antibodies from Santa Cruz biotechnologies are being used in thousands of publications annually. The company has been punished for their misconduct, and subsequently granted permission to produce antibodies from the relevant authorities again.

The following is the authors’ response to the original reviews.

**Public Reviews:**

**Reviewer #1 (Public Review):**
Summary:Despite the study being a collation of important results likely to have an overall positive effect on the field, methodological weaknesses and suboptimal use of statistics make it difficult to give confidence to the study's message.Strengths:Relevant human and mouse models approached with in vivo and in vitro techniques.Weaknesses:The methodology, statistics, reagents, analyses, and manuscripts' language all lack rigour.(1) The authors used statistics to generate P-values and Rsquare values to evaluate the strength of their findings.However, it is unclear how stats were used and/or whether stats were used correctly. For instance, the authors write: "Gaussian distribution of all numerical variables was evaluated by QQ plots". But why? For statistical tests that fall under the umbrella of General Linear Models (line ANOVA, t-tests, and correlations (Pearson's)), there are several assumptions that ought to be checked, including typically:(a) Gaussian distribution of residuals.(b) Homoskedasticity of the residuals.(c) Independence of Y, but that's assumed to be valid due to experimental design.So what is the point of evaluating the Gaussian distribution of the data themselves? It is not necessary. In this reviewer's opinion, it is irrelevant, not a good use of statistics, and we ought to be leading by example here.Additionally, it is not clear whether the homoscedasticity of the residuals was checked. Many of the data appear to have particularly heteroskedastic residuals. In many respects, homoscedasticity matters more than the normal distribution of the residuals. In Graphpad analyses if ANOVA is used but equal variances are assumed when variances among groups are unequal then standard deviations assigned in each group will be wrong and thus incorrect p values are being calculated.Based on the incomplete and/or wrong statistical analyses it is difficult to evaluate the study in greater depth.

We agree with the reviewer that we should lead by example and improve clarity on the use of the different statistical tests and their application. In response to the reviewer’s suggestion, we have extended the statistical section, focusing on the analyses used. Additionally, we have specified the statistical test used in the figure legends for each figure. Additionally, we did check for Gaussian distribution and homoskedasticity of residuals before conducting a general linear model test, and this has now been specified in the revised manuscript. In case the assumptions were not met, we have specified which non-parametric test we used. If the assumptions were not met, we specified which non-parametric test was used.

While on the subject of stats, it is worth mentioning this misuse of statistics in Figure 3D, where the authors added the Slc34a1 transcript levels from controls in the correlation analyses, thereby driving the intercept down. Without the Control data there does not appear to be a correlation between the Slc34a1 levels and tumor size.

We agree with the reviewer that a correlation analysis is inappropriate here and have removed this part of the figure.

There is more. The authors make statements e.g. in the figure levels as: "Correlations indicated by R2.". What does that mean? In a simple correlation, the P value is used to evaluate the strength of the slope being different from zero. The authors also give R2 values for the correlations but they do not provide R2 values for the other stats (like ANOVAs). Why not?

We agree with the reviewer and have replaced the R2 values with the Pearson correlation coefficient in combination with the P value.

(2) The authors used antibodies for immunos and WBs. I checked those antibodies online and it was concerning:(a) Many are discontinued.

Many of the antibodies we have used were from the major antibody provider Santa Cruz Biotechnology (SCBT). SCBT was involved in a scandal of animal cruelty and all their goats and rabbits were sacrificed, which explains why several antibodies were discontinued, while the mice antibodies were allowed to continue. This is unfortunate but out of our hands.

(b) Many are not validated.

We agree with the reviewer that antibody validation is essential. All antibodies used in this manuscript have been validated. The minimal validation has been to evaluate cellular expression in positive control tissue for instance bone, kidney, or mamma. Moreover, many of the antibodies have been used and validated in previous publications (doi: 10.1593/neo.121164, doi:10.1096/fj.202000061RR, doi: 10.1093/cvr/cvv187) including knockout models. Moreover, many antibodies but not all have been validated by western blot or *in situ* hybridization. We have included the following in the *Materials and Methods* section: “All antibodies used have been validated by testing in positive control tissue and/or Western blot or *in situ* hybridization”.

(c) Many performed poorly in the Immunos, e.g. FGF23, FGFR1, and Kotho are not really convincing. PO5F1 (gene: OCT4) is the one that looks convincing as it is expressed at the correct cell types.

We fail to understand the criticism raised by the reviewer regarding the specificity of these specific antibodies. We believe the FGF23 and Klotho antibodies are performing exceptionally well, and FGFR1 is abundantly expressed in many cell types in the testis. As illustrated in Figure 2E, the expression of Klotho, FGF23, and FGFR1 is very clear, specific, and convincing. FGF23 is not expressed in normal testis – which is in accordance with no RNA present there either. However, it is abundantly expressed in GCNIS where RNA is present. On the other hand, Klotho is abundantly expressed in germ cells from normal testis but not expressed in GCNIS.

(d) Others like NPT2A (product of gene SLC34A1) are equally unconvincing. Shouldn't the immuno show them to be in the plasma membrane?If there is some brown staining, this does not mean the antibodies are working. If your antibodies are not validated then you ought to omit the immunos from the manuscript.

We acknowledge your concerns regarding the NPT2A, NPT2B, and NPT2C staining. While the NPT2A antibody is performing well, we understand your reservations about the other antibodies. It's worth noting that NPT2A is not expressed in normal testis (no RNA either) but is expressed in GCNIS where the RNA is also present. Although it is typically present in the plasma membrane, cytoplasmic expression can be acceptable as membrane availability is crucial for regulating NPT2A function, particularly in the kidney where FGF23 controls membrane availability. We are currently involved in a comprehensive study exploring these phosphate transporters in the organs lining the male reproductive tract. In functional animal models, we have observed very specific staining with this NPT2A antibody following exposed to high phosphate or FGF23. Additionally, we are conducting Western Blot analyses with this antibody, which reinforces our belief that the antibody has a specific binding.

**Reviewer #2 (Public Review):**
Summary:This study set out to examine microlithiasis associated with an increased risk of testicular germ cell tumors (TGCT). This reviewer considers this to be an excellent study. It raises questions regarding exactly how aberrant Sertoli cell function could induce osteogenic-like differentiation of germ cells but then all research should raise more questions than it answers.Strengths:Data showing the link between a disruption in testicular mineral (phosphate)homeostasis, FGF23 expression, and Sertoli cell dysfunction, are compelling.Weaknesses:Not sure I see any weaknesses here, as this study advances this area of inquiry and ends with a hypothesis for future testing.

We thank the reviewer for the acknowledgment and highlighting that this is an important message that addresses several ways to develop testicular microlithiasis, which indicates that it is not only due to malignant disease but also frequent in benign conditions.

**Recommendations for the authors:**

**Reviewer #1 (Recommendations For The Authors):**
I applaud the authors' approach to nomenclature for rodent and human genes and proteins (italicised for genes, all caps for humans, capitalised only for rodents, etc), but the authors frequently got it wrong when referring to genes or proteins. A couple of examples include:(1) SLC34A1 (italics) refers to gene (correct use by the authors) but then again the authors use e.g. SLC34A1 (not italics) to refer to the protein product of SLC34A1(italics) gene. In fact, the protein product of the SLC34A1 (italics) gene is called NPT2A (non-italics).(2) OCT4 (italics) refers to gene (correct use by the authors) but then again the authors use e.g. OCT4 (not italics) to refer to the protein product of OCT4 (italics)gene. In fact, the protein product of the OCT4 gene (italics) gene is called PO5F1(non-italics).The problem with their incorrect and inconsistent nomenclature is widespread in the manuscript making further evaluation difficult.Please consult a reliable protein-based database like Uniprot to derive the correct protein names for the genes. You got NANOG correct though.

We thank the reviewer for addressing this important point. We have corrected the nomenclature throughout the manuscript as suggested.

(3) The authors use the word "may" too many times. Also often in conjunction with words like "indicates", and "suggests". Examples of phrases that reflect that the authors lack confidence in their own results, conclusions, and understanding of the literature are:"...which could indicate that the bone-specific RUNX2 isoform may also be expressed... ""...which indicates that the mature bone may have been..."Are we shielding ourselves from being wrong in the future because "may" also means "may not"? It is far more engaging to read statements that have a bit more tooth to them, and some assertion too. How about turning the above statements around, to :"...which shows that the bone-specific RUNX2 isoform is also expressed... ""...which reveals that the mature bone were..."...then revisit ambiguous language ("may", "might" "possibly", "could", "indicate" etc.) throughout the manuscript?It's OK to make a statement and be found wrong in the future. Being wrong is integral to Science.

Thank you for addressing this. We agree with the reviewer that it is fair to be more direct and have revised many of these vague phrases throughout the manuscript.

(4) The authors use the word "transporter" which in itself is confusing. For instance, is SLC34A1 an importer or an exporter of phosphate? Or both? Do SLC34As move phosphate in or out of the cells or cellular compartments? "Transporter" sounds too vague a word.

We understand that it might be easier for the reader with the term "importer". However, we should use the specific nomenclature or "wording" that applies to these transporters. The exact terminology is a co-transporter or sodium-dependent phosphate cotransporter as reported here (doi: 10.1152/physrev.00008.2019). Thus, we will use the terms “co-transporter” and “transporter” throughout the revised manuscript.